# Mapping Urban and Peri-Urban Land Cover in Zimbabwe: Challenges and Opportunities

Courage Kamusoko [1,*], Olivia Wadzanai Kamusoko [1,2], Enos Chikati [3] and Jonah Gamba [1]

1   LocaSense Research Systems, Harare, Zimbabwe; OliviaWadzanai.Kamusoko@locasensersys.com (O.W.K.); jonah.gamba@locasensersys.com (J.G.)
2   Department of Geography, Geospatial Sciences and Earth Observation, University of Zimbabwe, Mount Pleasant, Harare, Zimbabwe
3   Rainbow Secondary School, Gaborone, Botswana; chikati@rainbowschool.ac.bw
*   Correspondence: courage.kamusoko@locasensersys.com

**Abstract:** Accurate and current land cover information is required to develop strategies for sustainable development and to improve the quality of life in urban areas. This study presents an approach that combines multi-seasonal Sentinel-1 (S1) and Sentinel-2 (S2) data, and a random forest (RF) classifier in order to map land cover in four major urban centers in Zimbabwe. The specific objective of this study was to assess the potential of multi-seasonal (rainy, post-rainy, and dry season) S1, rainy season S2, post-rainy season, dry season S2, multi-seasonal S2, and multi-seasonal composite S1 and S2 data for mapping land cover in urban areas. The study results show that the combination of multi-seasonal S1 and S2 data improve land cover mapping in urban and peri-urban areas relative to only multi-seasonal S1, mono-seasonal S2, and multi-seasonal S2 data. The overall accuracy scores for the multi-seasonal S1 and S2 land cover maps are above 85% for all urban centers. Our results indicate that rainy and post-rainy S2 spectral bands, as well as dry-season S1 VV and VH bands (ascending orbit) are the most important features for land cover mapping. In particular, S1 data proved useful in separating built-up areas from cropland, which is usually problematic when only optical imagery is used in the study area. While there are notable improvements in land cover mapping, some challenges related to the S1 data analysis still remain. Nonetheless, our land cover mapping approach shows a potential to map land cover in other urban areas in Zimbabwe or in Sub-Sahara Africa. This is important given the urgent need for reliable geospatial information, which is required to implement the United Nations Sustainable Development Goals (UN SDGs) and United Nations New Urban Agenda (NUA) programmes.

**Keywords:** earth observation satellite; Sentinel-1; Sentinel-2; random forest; urban; peri-urban; Zimbabwe

## 1. Introduction

According to the United Nations Human Settlements Programme (UN-Habitat), 55% of the world's population now reside in urban areas, while in 2050 the urban population is expected to reach 68% [1,2]. About 90% of urban growth is expected to occur in less developed regions such as East Asia, South Asia, and Africa [1]. Recent reports indicate that Africa had the highest annual urbanization rates of 3.7% between 2010 and 2015, and 3.57% between 2015 and 2020 [1]. The rapid urbanization in Africa is attributed to natural increase, rural-urban migration, and the reclassification of rural areas as urban areas [1,3,4]. This has resulted in the increase of informal settlements within urban areas, and urban sprawl through peri-urbanization [4]. In some African countries, the physical extent of urban areas is growing much faster than their population and administrative boundaries [1]. Consequently, more land is taken for urban development. This has serious implications for climate change, greenhouse gas emissions, loss of biodiversity, environmental degradation, and energy consumption [5,6]. Given the rapid growth of urban population in Africa, local government authorities—responsible for urban planning and management—are failing to

provide adequate housing, basic services (provision of clean water and sanitation), and basic infrastructures such as transport and health facilities [1,7]. This is further worsened by the outbreak of epidemics and global pandemics such as COVID-19, which will impact more vulnerable citizens living in informal settlements [8].

In 2015, the global community adopted the 2030 United Nations Sustainable Development Agenda, which includes a specific Sustainable Development Goal (SDG) to "make cities and human settlements inclusive, safe, resilient and sustainable" [9]. In 2016, 167 countries formally agreed to the United Nations New Urban Agenda (NUA) that emphasizes the practical implementation of national urban policies and action plans [1]. In light of the 2030 Agenda for Sustainable Development and NUA, accurate and timely geospatial information and insights are required in order to implement practical sustainable urban development action plans. However, geospatial information on the urban and peri-urban extent is often not available or outdated in most countries in Sub-Sahara Africa [10]. While local government authorities in these countries recognize the importance of geospatial information for sustainable urban planning and development, efforts to produce new or update geospatial information have been constrained by poor funding, as well as the high cost of conducting conventional land use surveys and aerial photography [11]. It is also noteworthy that most government policy makers do not prioritize investing in geospatial technology and information despite its contribution to spatial urban planning in particular and sustainable development in general [12]. This is partly attributed to the lack of communication among national and local government authorities, urban planners, politicians, and citizens [3]. Furthermore, "donor fatigue" has resulted in the decline of official development assistance (ODA) funding for mapping projects. Consequently, local government authorities or mapping agencies in most Sub-Sahara African countries fail to produce timely, reliable, and accurate geospatial information.

Remotely-sensed satellite data can be used to produce critical geospatial information at a regional scale since data can be obtained in a cost-effective manner and within a short acquisition time [13]. The literature review shows many studies that have used remotely-sensed satellite data for mapping and monitoring land cover changes [14]. This is due to the fact that moderate to high spatial resolution remotely-sensed satellite data (10–50 m) such as the Landsat series, Sentinel-1 and Sentinel-2, have a relatively good global coverage and are available free of charge. Furthermore, the advancement in machine learning methods such as support vector machines [15,16], random forests [17], and deep learning [18] has also increased land cover mapping and monitoring studies. However, most urban land cover mapping studies have focused on megacities and large urban centers located in China, the US, and Europe [19,20]. Although megacities and large cities are massive centers of economic activities, the fastest growing cities are the small and medium cities with less than 1 million inhabitants [4,21]. According to the United Nations, small and medium cities account for about 59% of the global urban population. To date, most of these cities are still poorly quantified, particularly in Sub-Sahara Africa. As a result, urban land cover information is still sparse in small and medium cities save for a case-study analysis of individual cities [10,13,22–30].

While remotely-sensed satellite data have been used successfully to map urban land cover, major challenges still remain [13,31]. This is attributed to the highly heterogeneous nature of urban areas [19], as well as the spectral similarity between sparse and fragmented built-up areas and other land cover types in peri-urban areas [32]. Previous studies show that built-up areas in sparse urban or peri-urban areas appear identical to fallow cropland and bare areas given that these features exhibit high reflectance in the visible-infrared wavelengths [6,10]. For example, the spectral similarity between newly-developed peri-urban areas and non-urban surfaces such as fallow cropland fields and bare areas has been reported to be problematic, especially with moderate spatial resolution satellite imagery in Harare, Zimbabwe [25]. In addition, mapping land cover in peri-urban areas is notoriously difficult since some built-up areas are made of the same materials found in the surrounding areas [10], which results in low accuracy due to the low object-to-background contrast [33].

The availability of synthetic aperture radar (SAR) Sentinel-1 (S1) and optical Sentinel-2 (S2) data provide a great opportunity to address some of the key challenges. This is due to the fact that S1 and S2 data have high spatial and temporal resolutions. Although previous studies have shown the utility of S2 for urban land cover mapping [34–36], few studies have combined S1 and S2 for land cover mapping in urban areas [31], particularly in Sub-Sahara Africa. The primary goal of this study was to map land cover in four major urban centers in Zimbabwe using multi-seasonal S1 and S2 data and a random forest (RF) classifier. The specific objective of this study was to assess the potential of multi-seasonal (rainy, post-rainy, and dry season) S1 data, rainy season S2 data, post-rainy season S2 data, dry season S2 data, multi-seasonal S2 data, and multi-seasonal S1 and S2 data for mapping land cover in major urban areas in Zimbabwe. The central premise of our approach is that spectral-temporal features derived from multi-seasonal S2 data can help discriminate the built-up areas from cropland and bare areas, while multi-seasonal S1 data can help detect the built-up areas given the high backscatter of some man-made objects. We used the RF classifier since previous studies have shown the method to be effective in a similar landscape [37]. The remaining parts of this paper are organized as follows. Section 2 introduces the study area, while Section 3 provides details on the land cover mapping methodology. The results are presented in Section 4, while discussions are provided in Section 5. Finally, conclusions are presented in Section 6.

## 2. Study Area

According to the Zimbabwe National Statistical Agency [38], "urban" refers to a designated urban area with a compact settlement pattern with more than 2500 inhabitants, of which 50% are employed in the non-agricultural sector. Zimbabwe's urban landscape encompasses metropolitan areas of Harare and Bulawayo, cities or municipalities, towns, and as many as 472 small urban centers in the form of "growth points", district service and rural service centers [39,40]. The urban population in Zimbabwe grew from 677,270 in 1962 to 3,409,848 inhabitants in 2012. This represents a 20% increase in urban population in the country. According to ZimStats [38], 33% of the population in Zimbabwe is urban. Currently, the total population of Harare and Bulawayo accounts for about 62% of the total urban population in Zimbabwe [38].

Four major urban centers—Harare metropolitan area, Bulawayo, Mutare, and Gweru—with a population above 100,000 people were selected as case studies (Figure 1). The four major urban centers are characterized by rapid urban development, differences in physical and socio-economic geography, and climate. Furthermore, some of the urban centers are experiencing rapid informal and unplanned developments, particularly in peri-urban areas. This makes it challenging to accurately map land cover in these urban landscapes. Therefore, the four major urban centers in Zimbabwe serve as important case studies to test our land cover mapping approach. In addition, this study is shifting focus from a single to multiple case study approach in order to understand how mono-seasonal or multi-seasonal optical and SAR data improve land cover mapping in different urban landscapes.

The Harare metropolitan area comprises the City of Harare—which is the capital and largest city in Zimbabwe, Chitungwiza municipality, Ruwa, and Epworth Local Boards. The metropolitan area is characterized by a warm, rainy season from November to March, a cool, dry season from April to August, and a hot, dry season in October. Daily temperatures range from about 7 to 20 °C in July (coldest month), and from 13 to 28 °C in October (hottest month). The metropolitan area receives a mean annual rainfall ranging from 470 to 1350 mm between November and March. The population in the City of Harare increased from approximately 310,360 in 1962 to 1,435,784 in 2012, while the population of Chitungwiza grew from approximately 14,970 in 1969 to 354,472 in 2012 [41]. The population of Epworth increased from 114,067 in 2002 to 161,840 in 2012 [38]. The Harare metropolitan area represents over 47% of the total urban population in Zimbabwe. The City of Harare has many major primary and secondary industries, while Chitungwiza has a small industrial park.

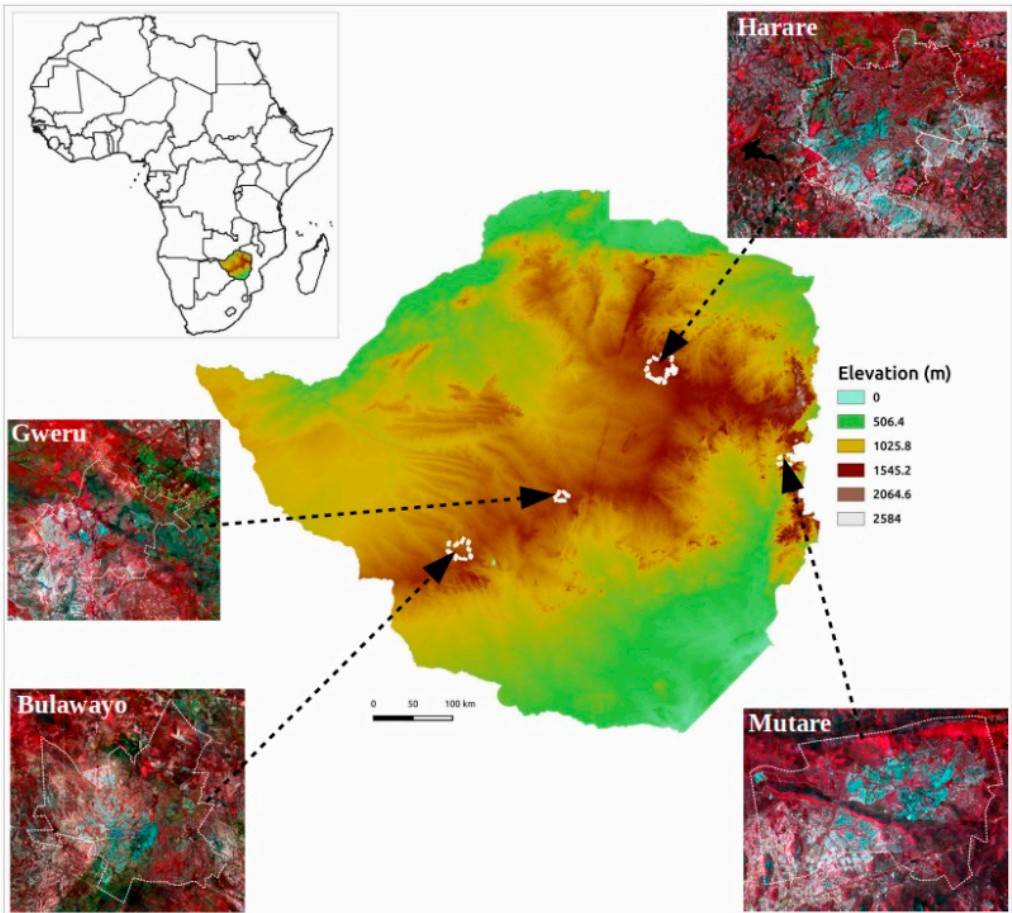

**Figure 1.** Location of the four major urban centers.

Bulawayo is the second largest city in Zimbabwe, which is located in the southwestern part of Zimbabwe. The modern Bulawayo city was founded in 1893 and attained city status in 1943. Bulawayo is characterized by a dry, cool winter season from May to July, a hot, dry period from late August to early November, and a warm, rainy period from early November to March. The hottest month is October, while the coldest is July. The temperatures range from an average of 21 °C in July to 30 °C in October, while the annual rainfall ranges between 588 and 600 mm. The population of the city increased significantly from 210,620 in 1969 to 676,650 in 2002, and then decreased to 655,675 in 2012 [38,41]. Bulawayo is the manufacturing and industrial center of Zimbabwe with a large presence of heavy industries. However, much of the industrial infrastructure has deteriorated during the past years due to the poor economic environment in Zimbabwe.

Mutare is located in the eastern part of Zimbabwe. The city was founded in 1897 as a fort. Mutare has a humid subtropical climate, which is characterized by a dry, cool winter season from May to July, a hot, dry period from late August to early November, and a warm, rainy period from early November to March. The coldest month is July (minimum 6 °C and maximum 20 °C), while the hottest month is October (minimum 16 °C and maximum 32 °C). The annual rainfall is about 818 mm, which falls mostly from November to March. The population increased significantly from 42,540 in 1969 to 188,243 in 2012 [38,41]. The main economic activities in Mutare are citrus farming, mining, and forestry. Two of the largest food producers in Zimbabwe, Cairns Foods and Tanganda Tea, are located in Mutare. Mining activities include gold and diamonds, as well as gravel quarries around the city. Forestry companies such as The Wattle Company, Allied Timbers, and Border Timbers are also located in Mutare. More importantly, Mutare is an important gateway to the sea since it is located 290 km from the port of Beira in Mozambique.

Gweru is located about 285 km south of the Harare metropolitan area. The city is characterized by a dry, cool winter season from May to July, a hot, dry period in August to early November, and a warm, rainy period from early November to April. The hottest month is October, while the coldest is July. The temperatures range from an average of 21 °C in July to 30 °C in October, while the annual rainfall is about 684 mm. The population increased significantly from 38,480 in 1969 to 158,233 in 2012 [38,41]. Gweru is centrally located between Harare and Bulawayo, and therefore is an important transport hub. The city provides services for mining and commercial agriculture activities in the surrounding areas. Gweru also produces ferrochromium, textiles, dairy foods, footwear, and building materials.

## 3. Methods

The methodology used in this study comprises data preparation and land cover mapping. The following subsections describe satellite imagery, reference datasets, and land cover mapping procedures.

### 3.1. Data Preparation

3.1.1. Satellite Imagery

We derived seasonal Sentinel-1 (S1) and Sentinel-2 (S2) data for 2020 from the Google Earth Engine [42] to map land cover in four major urban areas in Zimbabwe (Table 1). The seasonal S1 data consist of mean and median rainy season S1, mean and median post-rainy season S1, and mean and median dry season S1 composites. We used mean and median seasonal S1 data since the imagery shows a lower speckle than the single-date imagery. As a result, we did not perform speckle reduction, which generally reduces spatial resolution. The seasonal S2 data comprise median rainy season S2, median post-rainy season S2, and median dry season S2 composites (Table 1). The rainy season is between January and March, the post-rainy is between April and June, and the dry season is between July and October in the study area. We used multi-seasonal S2 data since previous studies revealed that multi-seasonal optical data are useful for identifying phenological changes [43] (Figures S1–S5).

**Table 1.** Summary of Sentinel-1 (S1) and Sentinel-2 (S2) data used in the study.

| Compiled Imagery | Date Range | Season | Number of Images/Bands | Remarks |
|---|---|---|---|---|
| Mean and median S1 | 1 January–30 March 2020 | Rainy | 4 | IW swath mode 250 km, VV and VH polarization, pixel spacing 10 m, Ascending orbit |
| | 1 April–30 June 2020 | Post-rainy | 4 | |
| | 1 July–30 October 2020 | Dry | 4 | |
| Median S2 | 1 January–30 March 2020 | Rainy | 9 | Bands 2, 3, 4, 8, at 10 m spatial resolution; Bands 5, 6, 7, 8a, 11, and 12 resampled to 10 m |
| | 1 April–30 June 2020 | Post-rainy | 9 | |
| | 1 July–30 October 2020 | Dry | 9 | |

S1 and S2 data are derived from a constellation of satellites developed by the European Space Agency (ESA) under the Copernicus program [44]. The S1 mission comprises a constellation of Sentinel-1A and Sentinel-1B satellites, which provides a 12-day (ground track) repeat cycle for one satellite, and a 6-day (ground track) repeat cycle for two satellites [45]. The Sentinel-1 constellation provides C-band (5.6 cm) synthetic aperture radar (SAR) acquired in different modes [44]. In this paper, we used the VV and VH interferometric wide swath (IW) and ground range detected (GRD) S1 data, which have been processed and terrain corrected. Co-polarization VV refers to the vertical transmit and vertical receive, while cross-polarization VH refers to the vertical transmit and horizontal receive [46]. Sentinel-2 is a wide-swath, high-resolution, multispectral imaging mission with a global 5-day revisit frequency [44]. The Sentinel-2 multispectral instrument (MSI) provides 13 spectral bands [44]. In this study, we selected nine spectral bands from S2 level-2A orthorectified atmospherically corrected surface reflectance imagery, which are commonly used for land cover mapping applications. The selected bands are band 2 (Blue), band 3 (Green), band 4 (Red), band 5 (Vegetation red edge) (VRE1), band 6 (VRE2), band 7 (VRE3), band 8 (Near infrared) (NIR), band 11 (Short-wave infrared) (SWIR1), and band 12 (SWIR2).

Figure 2 shows the multi-seasonal S1 false color composite (ascending orbit) and S2 false color composite imagery for Harare. Note that the S1 imagery is displayed in false color VV (red), VH (green), and VV (blue) for visualization purposes only, while the S2 imagery is displayed in false color band 8 (red), band 4 (green), and band 3 (blue). The black square inset in Figure 2 shows a newly-developing peri-urban area in the south-western part of Harare, which is used to extract subset images for better visualization. Figure 3 shows S1 rainy season (S1RS), S1 post-rainy season (S1PS), S1 dry season (S2DS), rainy season (S2RS), S2 post-rainy season (S2PS), S2 dry season (S2DS) image subsets, as well as Google Satellite image subsets. The visual analysis of both S1 and S2 reveals that the vegetation cover decreases from the rainy to dry season, while open or bare areas increase during the same period. Figure 2a–c shows that the S1 composite imagery is generally characterized by a combination of surface, volume, and double-bounce scattering. For example, surface scattering is mainly comprised of low-vegetated cropland and bare soils, as well as roads and other paved surfaces which are shown in blue. However, volume scattering (which is shown in green) is dominated by vegetation canopy, as well as some built-up surfaces. Finally, double-bounce scattering (in pink) is mainly from buildings and other man-made structures oriented towards the SAR look direction (Figure 3). Figure 4 also shows seasonal differences in both the S1 and S2 imagery for Bulawayo.

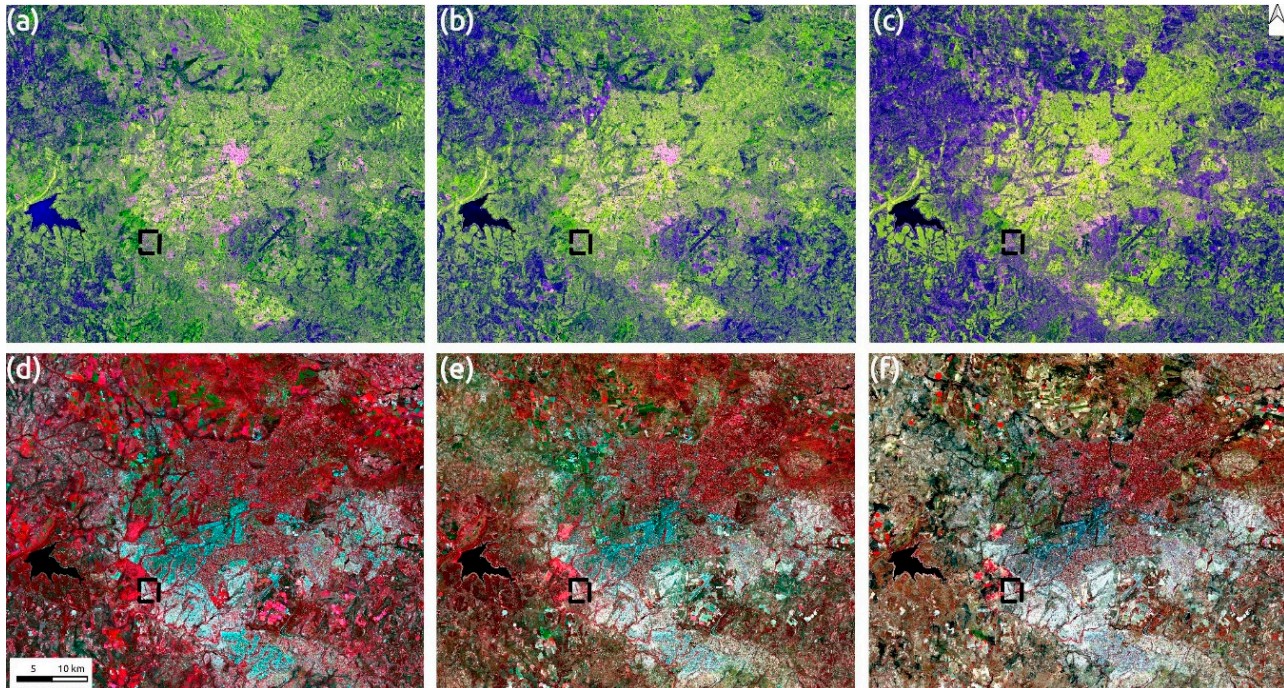

**Figure 2.** Sentinel imagery for Harare: (**a**) S1 rainy season (S1RS); (**b**) S1 post-rainy season (S1PS); (**c**) S1 dry season (S1DS); (**d**) S2 rainy season (S2RS); (**e**) S2 post-rainy season (S2PS); and (**f**) S2 dry season (S2DS). S1 imagery is displayed in false color red (R), green (G), blue (B) (VV, VH, VV), while S2 imagery is displayed in false color RGB (8,4,3). Note VV refers to vertical transmit and vertical receive; and VH refers to vertical transmit and horizontal receive.

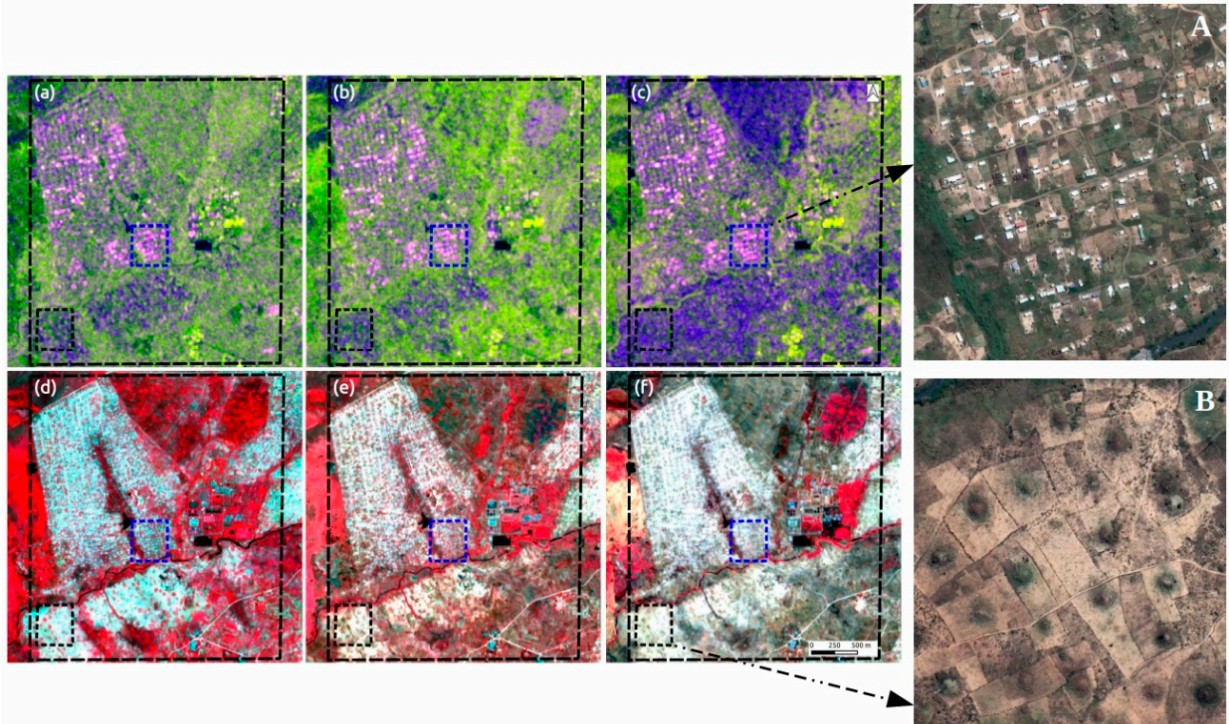

**Figure 3.** Newly-developing peri-urban area: (**a**) S1RS; (**b**) S1PS; (**c**) S1DS; (**d**) S2RS; (**e**) S2PS; (**f**) S2DS subsets, and Google Satellite imagery (locations (**A**,**B**)). The blue rectangle shows sparse built-up areas in newly developed peri-urban areas, while the black rectangle shows typical cropland areas. S1 imagery is displayed in false color RGB (VV, VH, VV), while S2 imagery is displayed in false color RGB (8,4,3).

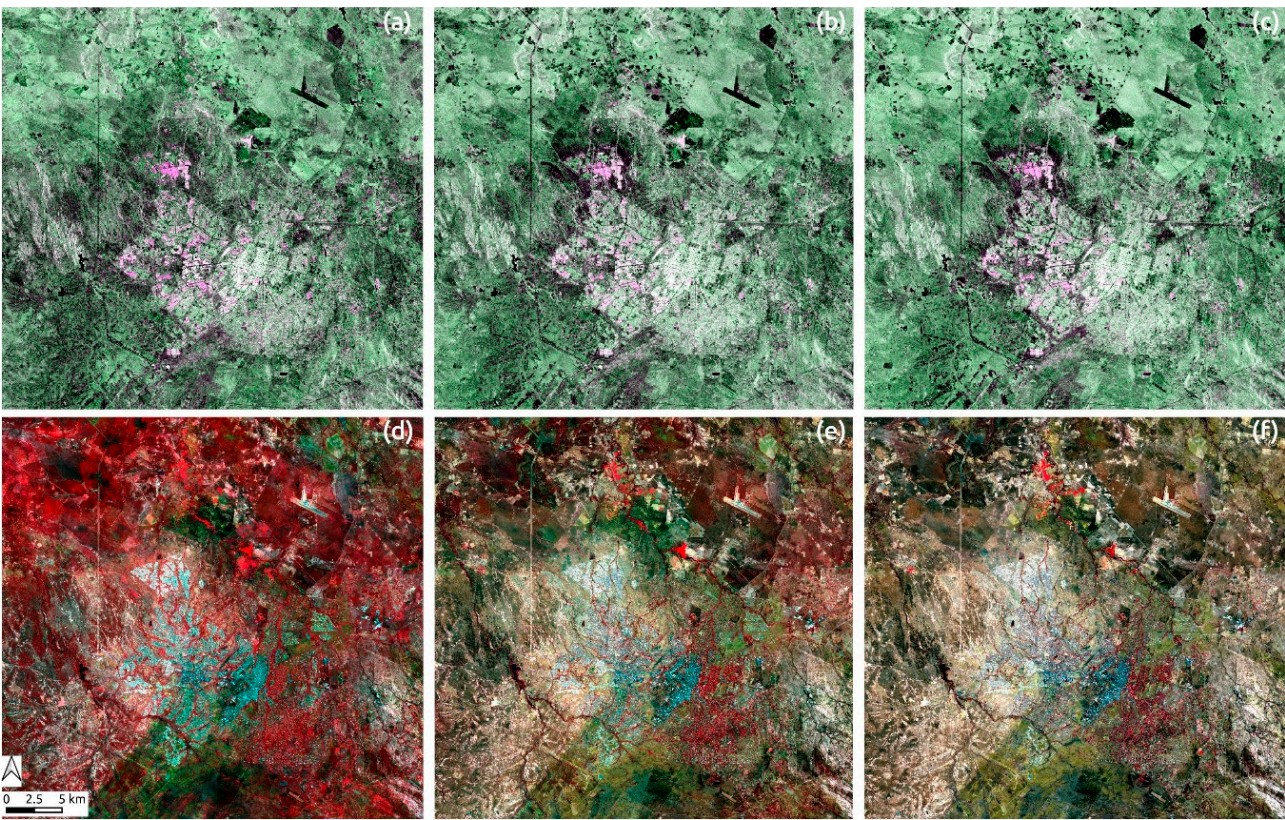

**Figure 4.** Sentinel imagery for Bulawayo: (**a**) S1RS; (**b**) S1PS; (**c**) S1DS; (**d**) (S2RS); (**e**) S2PS; and (**f**) S2DS. S1 imagery is displayed in false color RGB (VV, VH, VV), while S2 imagery is displayed in false color RGB (8,4,3).

### 3.1.2. Reference Data for Land Cover Classification

Reference data for training and testing were generated from a variety of sources. For the Harare metropolitan area, reference polygons were digitized from a very high resolution imagery available in Google Earth™, and digital orthophotos which were obtained from the Department of the Surveyor-General (DSG). The digital orthophotos were processed from digital aerial photographs which were acquired by DSG and the Japan International Cooperation Agency (JICA) in 2015 during the "The Development of Geospatial Information Database Project" in Zimbabwe. The reference datasets were originally compiled for a pilot urban land use project in 2012–2013 [25], which focused on the classification of built-up and non-built up areas. Therefore, there is more reference data on built-up areas than other land cover classes. The reference data for Bulawayo, Mutare, and Gweru were digitized from the very high resolution imagery available from Google Earth™. While a lot of effort was made to prepare reliable and accurate reference data, it should be noted that reference data were compiled from different sources. Therefore, it is inevitable that some errors are found within the reference data. Nonetheless, the reference data are very useful since some locations, especially in the peri-urban areas were carefully interpreted on digital orthophotos and checked during the fieldwork in Harare in 2019 and 2020.

Land cover is the observed biophysical cover on the earth's surface, while land use refers to the human-environment interaction and is characterized by human activities. For example, built-up is a land cover, while the high density residential area is a land use. In this study, the focus is on mapping land cover. Table 2 shows the target land cover classes, which are based on the "Forestry Commission (Zimbabwe) and DSG national woody cover classes" classification schemes and the author's a priori knowledge of the study areas. The original land cover classes were modified with the aid of orthophotos and fieldwork.

In total, six land cover classes were considered in this study: (1) Built-up; (2) bare areas; (3) cropland; (4) woodland; (5) grass/open areas; and (6) water.

**Table 2.** Land cover classification scheme and distribution of training polygons.

| Land Cover | Description | Training Sites Per Class | | | |
|---|---|---|---|---|---|
| | | Harare | Bulawayo | Mutare | Gweru |
| Built-up (BU) | Residential, commercial, services, industrial, transportation, communication, and utilities and construction sites. | 2113 | 806 | 419 | 464 |
| Bare areas (BA) | Bare sparsely vegetated area with >60% soil background. Includes sand and gravel mining pits, rock outcrops. | 1091 | 139 | 152 | 111 |
| Cropland (Cr) | Cultivated land or cropland under preparation, fallow cropland, and cropland under irrigation. | 1008 | 147 | 145 | 130 |
| Woodland (Wd) | Woodlands, riverine vegetation, shrub and bush. | 331 | 328 | 59 | 52 |
| Grass/open areas (Gr) | Grass cover, open grass areas, golf courses, and parks. | 1095 | 434 | 205 | 277 |
| Water (Wt) | Rivers, reservoirs, and lakes. | 73 | 18 | 7 | 23 |

### 3.1.3. Land Cover Mapping Approach

We used the random forest (RF) classifier which is wrapped in the RStoolbox package in R [47,48] to classify multi-seasonal S1 (SS1), S2 rainy season (S2RS), S2 post-rainy season (S2PS), S2 dry season (S2DS), multi-seasonal S2 (SS2), and multi-seasonal composite S1 and S2 (SC) data. The SS1 data composite comprises S1 rainy season (S1RS), S1 post-rainy season (S1PS), S1 dry season (S1DS) data, while the SS2 data composite comprises S2RS, S2PS, and S2DS data (Table 1). The SC combines all the rainy, post-rainy, and dry season S1 and S2 data.

The RF classifier uses bootstrap sampling to build many single decision tree models [17,49,50]. In general, a random subset of predictor variables or bands is used to split an observation data into homogeneous subsets. The subsets are used to build each decision tree model and a prediction [50,51]. Then, single decision tree model predictions are averaged in order to produce the final labeling [52]. The out-of-bag (OOB) sample data are used to evaluate performance, while importance measures are computed based on the proportion between misclassifications and the OOB sample [53]. This provides an unbiased estimation of the generalization error which is used for feature selection [17]. The advantages of the RF classifier are: (i) they can handle both numerical and categorical variables, (ii) are free from normal distribution assumptions, and (iii) are relatively robust to outliers and noise [17,50,53].

In this study, 70% of the reference dataset were used for training, while 30% were used for testing. A training model was set up to find the optimal model parameters, as well as to check the initial model performance based on repeated cross-validation. A total of 500 decision trees were used in the RF model. We used a tune length of 3 (which defines the number of levels for each tuning parameter) and a cross-validation of 5 (which represents the number of cross-validation resamples during model tuning). The performance of held-out samples was calculated and then the model with the optimal resampling statistic was selected. We also defined a test model in order to evaluate the performance of the RF classifier. After evaluating the test model, we applied a majority filter based on a

$3 \times 3$ window filter in order to remove the small pixels that cause a salt and pepper effect on all land cover maps. We also computed feature importance scores using the mean decrease accuracy.

## 4. Results

### 4.1. Land Cover Mapping and Analysis for Harare

4.1.1. Evaluation of the Training and Test Models

First, we evaluated the training and test models for Harare since the study area has a comprehensive reference data. To assess the RF model, the out-of-bag (OOB) estimate of the error rate, overall accuracy, and class errors were estimated using a five-fold cross-validation. Table 3a shows a summary of multi-seasonal S1 (SS1), S2 rainy season (S2RS), S2 post-rainy season (S2PS), S2 dry season (S2DS), multi-seasonal S2 (SS2), and multi-seasonal composite S1 and S2 (SC) training models. The SS1 training model has the worst performance as shown by the highest OOB error rate and relatively moderate training accuracy. In general, marginal differences are observed for the S2RS, S2PS, and S2DS training models in terms of the OOB error rate and training accuracy. Note that the OOB error rate slightly increases from rainy season to dry season, while training accuracy slightly increases from dry season to rainy season. This indicates that training models based on mono-seasonal (single season) S2 data have a relatively similar model performance, and thus will produce a less than optimum classification accuracy. However, there is a significant decrease in the OOB error rate and a subsequent increase in the training accuracy for the SS2 and SC training models. This indicates an improved model performance and hence, a likelihood of better classification accuracy.

**Table 3.** (**a**) Summary of SS1; S2RS; S2PS; S2DS; SS2; and multi-seasonal composite S1 and S2 (SC) training model results. (**b**) Summary of accuracy results (%) for SS1; S2RS; S2PS; S2DS; SS2; and SC test models.

| (a) | | | | | | |
|---|---|---|---|---|---|---|
| **Component** | **SS1** | **S2RS** | **S2PS** | **S2DS** | **SS2** | **SC** |
| No. of variables (bands) used | 12 | 9 | 9 | 9 | 27 | 39 |
| No. of variables tried at each split | 2 | 2 | 5 | 2 | 14 | 2 |
| OOB estimate of error rate | 29.3% | 14.4% | 15.4% | 15.8% | 8.3% | 7.6% |
| Training accuracy | 70.5% | 84.8% | 83.9% | 83.8% | 91% | 92.1% |

| (b) | | | | | | | | | | | |
|---|---|---|---|---|---|---|---|---|---|---|---|
| **Class** | **SS1** | | **S2RS** | | **S2PS** | | **S2DS** | | **SS2** | | **SS1&2** | |
| | **PA** | **UA** | **PA** | **UA** | **PA** | **UA** | **PA** | **UA** | **PA** | **UA** | **PA** | **UA** |
| Built-up | 58.3 | 58.9 | 83.9 | 76.7 | 84.4 | 72.5 | 80.9 | 64.9 | 88.9 | 77.1 | 93.7 | 79.4 |
| Bare areas | 54.1 | 70.8 | 64.4 | 74.7 | 64.2 | 75.9 | 57.9 | 70.6 | 66.4 | 79.3 | 66.1 | 83.9 |
| Cropland | 72.4 | 72.1 | 79.7 | 73 | 73.5 | 57.3 | 75.8 | 63.1 | 86.1 | 71.6 | 89.3 | 75.9 |
| Woodlands | 70.8 | 61.8 | 95.7 | 94.1 | 91.7 | 93.7 | 94.9 | 88.2 | 97 | 98.3 | 98.8 | 98.7 |
| Grass/open areas | 59.1 | 54.4 | 72 | 77.5 | 37.8 | 57.1 | 40.8 | 62.1 | 64.7 | 82.9 | 70.8 | 84.9 |
| Water | 98.8 | 99 | 99.6 | 99.7 | 100 | 95.9 | 98.1 | 99.5 | 100 | 98 | 100 | 99 |
| Overall accuracy | 68.7 | | 82.1 | | 74.7 | | 74.3 | | 83.6 | | 86.2 | |
| 95% CI | 67.5–69.8 | | 81.2–83.1 | | 73.7–75.8 | | 73.2–75.3 | | 82.7–84.5 | | 85.4–87.1 | |

Note: PA: Producer's accuracy; UA: User's accuracy; CI: Confidence interval.

In general, SS2 and SC training models have the lowest class errors, while the SS1 model has the highest class errors (Figure 5). However, all training models exhibit relatively high errors for the bare areas and grass/open classes. The cross-validation results reveal three important insights regarding the model training performance. First, both SS2 and SC training models outperformed the other training models as shown by the OOB errors, overall accuracy, and class errors. Second, S2RS, S2PS, and S2DS training models have relatively high errors for the bare areas, grass/open areas, cropland, and built-up classes. This suggests that mono-seasonal S2 data are not optimal for land cover mapping in the study area. Third, the SS1 training model has the worst performance, indicating serious classification problems. Therefore, multi-seasonal S1 data are not suitable for land cover mapping in the study area.

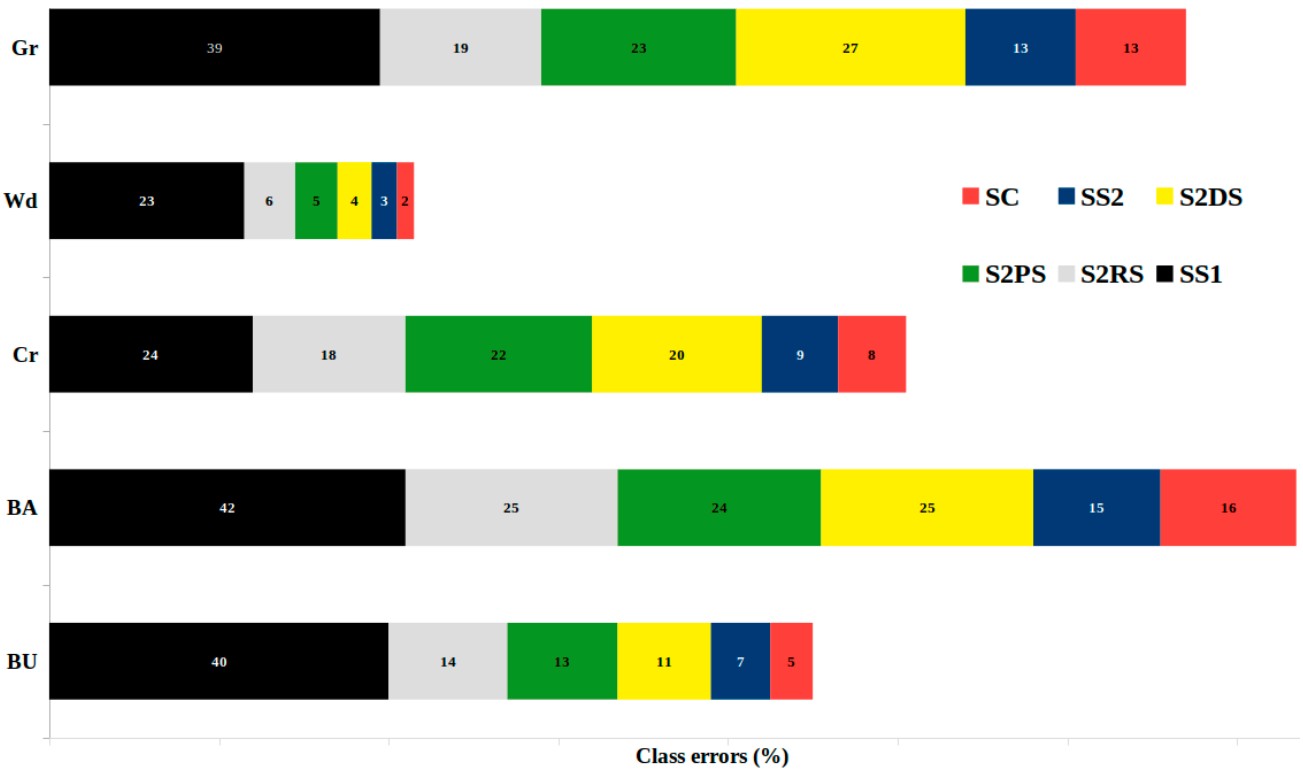

**Figure 5.** Training model land cover class errors for Harare.

Table 3b shows the overall accuracy and individual class accuracy results for all test models. The SS1 test model has the lowest overall accuracy, while the S2PS and S2DS test models have a moderate overall accuracy. Generally, the SC test model has the highest overall accuracy followed by S2RS and SS2 test models. Individual land cover accuracies exhibited a high variability across different test models. With regard to the built-up class, the producer's accuracy score is consistently higher than the user's accuracy score for all test models with the exception of the SS1 test model. The S2RS, S2PS, S2DS, SS2, and SC test models have high errors of commission, which suggest an overestimation of the built-up class. However, the relatively low producer's and user's accuracy for the SS1 test model indicates a severe underestimation and overestimation of the built-up class. For the bare areas class, the user's accuracy is higher than the producer's accuracy for all models indicating high errors of omission. In contrast, the cropland class has the high producer's accuracy and hence, low errors of omission. With respect to the woodland class, the SS1 model has the lowest user's accuracy, which indicates high errors of commission and thus, an overestimation of this class. Both the S2PS and S2DS models have the low producer's accuracy for the grass/open areas, which indicates a gross underestimation of this class. This suggests that the use of individual post-rainy and dry season S2 data is insufficient to

extract grass/open areas in the study area. The water class has relatively high individual accuracy scores indicating a good model performance for this class.

4.1.2. Evaluation of Land Cover Maps

In general, severe classification problems are observed in the SS1 land cover map (Figure 6a). An overestimation of built-up areas is observed, particularly in the far northern part of the study area, while in some cases built-up areas are completely omitted or misclassified as woodland areas. The analysis of the SS1 land cover reveals four important insights with regards to the land cover classification based on multi-seasonal S1 data. First, built-up areas are underestimated or completely omitted in developing peri-urban areas, which are mainly new housing development areas (see location A in Figure 6b). This is due to the fact that built-up areas which are not oriented orthogonal to the S1 sensor look direction are not captured in the S1 imagery since the incident SAR beam is reflected away from the sensor. Second, some built-up areas in core developed urban settlements which are not oriented orthogonal to the S1 sensor look direction are misclassified as woodland areas (see location B in Figure 6b). However, woodlands on hills or mountain slopes which are oriented in the north-east to south-west direction are misclassified as built-up areas. Location C (Figure 6b) shows that woodland areas which are located on slopes or hills appear similar to the built-up areas in developed urban areas. This is the main cause of confusion between built-up and woodland areas [54]. Third, built-up areas in developing peri-urban and developed core urban areas which are oriented from the north-west to south-east direction are correctly classified (see location D in Figure 6b). This is due to the double bounce and "cardinal" effects, which occur when man-made structures are orthogonal to the SAR illumination direction [33,55–57]. As a result, the built-up areas appear brighter due to a strong SAR backscatter (Figure 6b). Fourth, it is important to note that cropland areas are not confused with built-up areas. This is very significant since the spectral confusion between the cropland and built-up areas in optical imagery is usually one of the main causes of a poor classification accuracy. Figure 6c shows box plots of backscatter (dB) derived from the training data. The overlap between most of the classes is clearly observed from the distributions, which vary significantly. While there is a high backscattering response in the VV polarization for the built-up class, many outliers are present in this class (Figure 6c). This indicates that there is a significant confusion between the built-up and woodland areas.

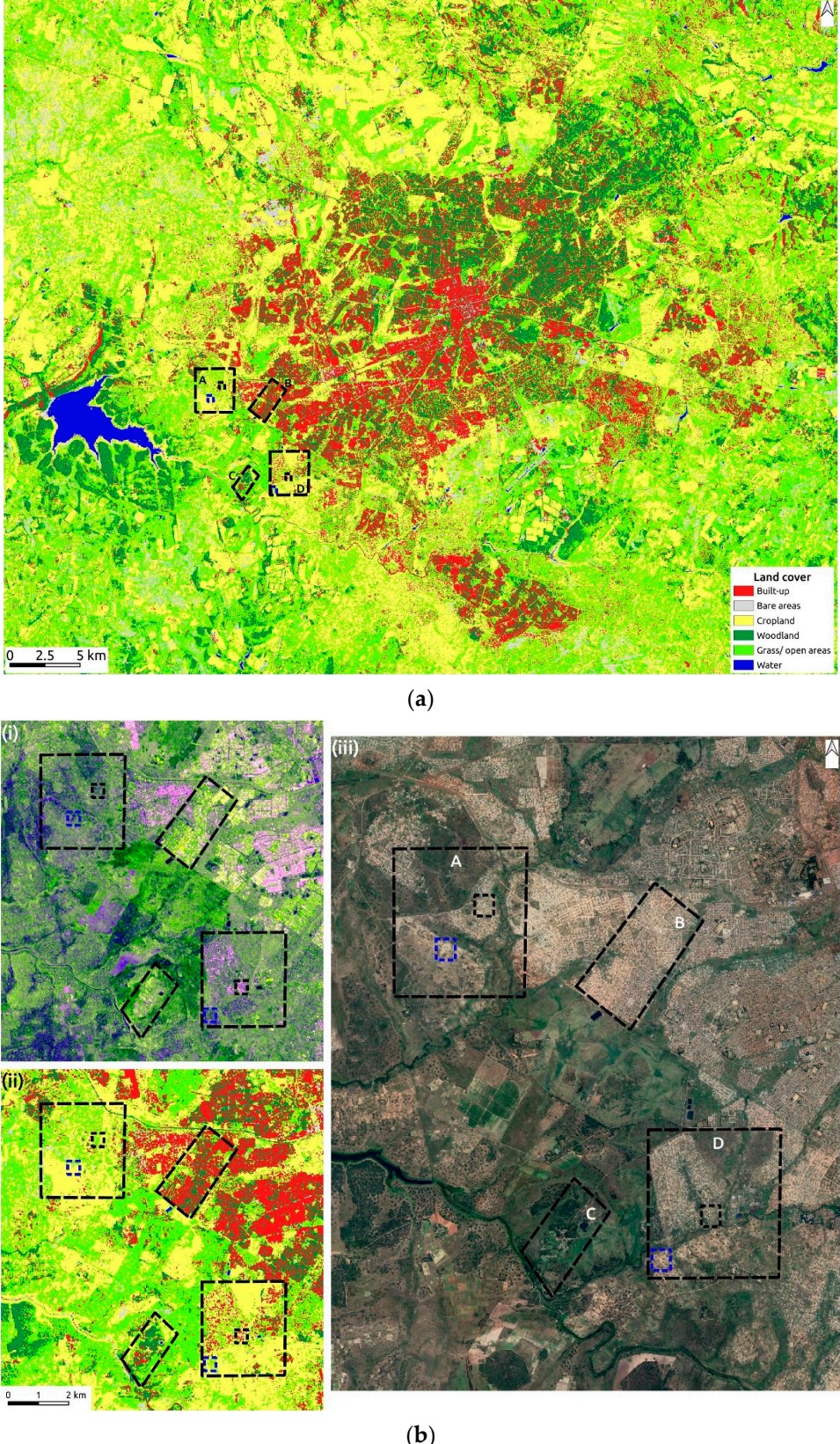

(**a**)

(**b**)

**Figure 6.** *Cont.*

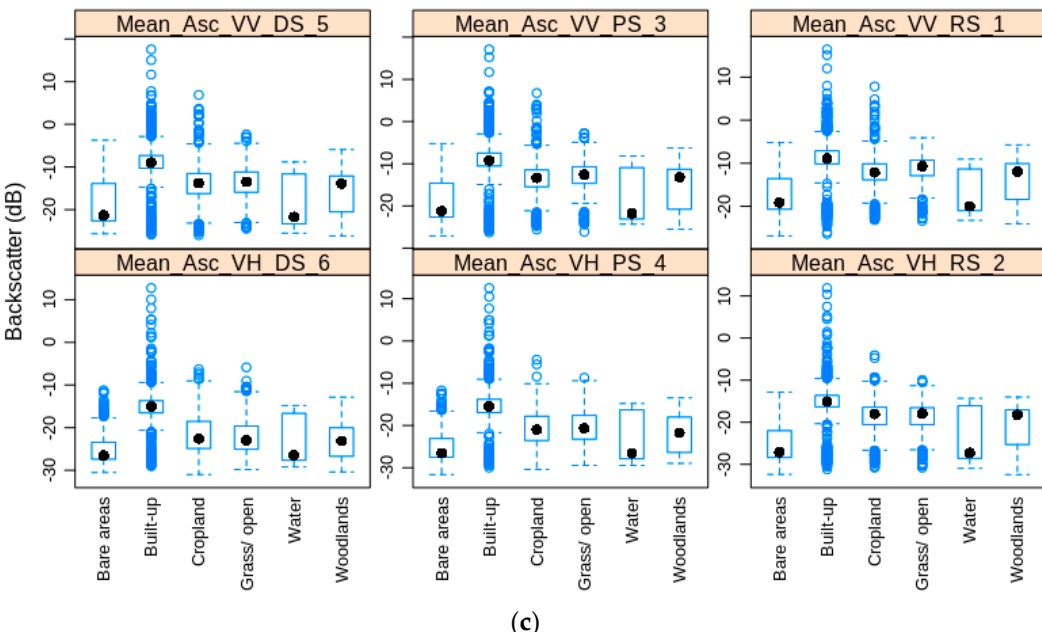

**(c)**

**Figure 6.** (**a**) SS1 land cover map: (A) Developing peri-urban; (B) developed urban area; (C) woodland area on a hill; and (D) developing peri-urban area. (**b**) (i) Seasonal S1 (SS1) imagery in false color, (ii) SS1 land cover map, (iii) Google Satellite imagery. (**c**) Box plots of the S1 intensities and land cover classes. Note that Mean_Asc_VV_DS_5 refers to the mean ascending VV dry season.

The rainy season (S2RS) land cover map shows many classification errors due to an overestimation of built-up areas (Table 3b) and spectral confusion between the built-up and cropland or bare areas (Figure 7a). In the S2 rainy season (S2RS) imagery fields planted with crops such as maize or other crop covers are distinguishable from the built-up areas or bare rock outcrops. However, it is difficult to separate cropland areas where land is being prepared for cultivation (bare cropland areas) from built-up areas in the S2RS imagery, especially in developing peri-urban areas (Figure 3d). Locations A to D, in Figure 7b, illustrate the misclassification of cropland as built-up areas. It is also noteworthy to point out that built-up areas in densely-vegetated low density residential areas (located to the north and north-east of the study area) are underestimated. This is due to the fact that most of the houses in these locations are partially or totally obscured by trees, which are leaf-on during the rainy season. In addition, grass/open areas are underestimated since these areas appear spectrally similar to the cropland areas during the rainy season (Figure 7c). In the study area, most grass/open areas have an irregular pattern, while the cropland areas are composed of regular and homogeneous patches which are relatively easy to detect.

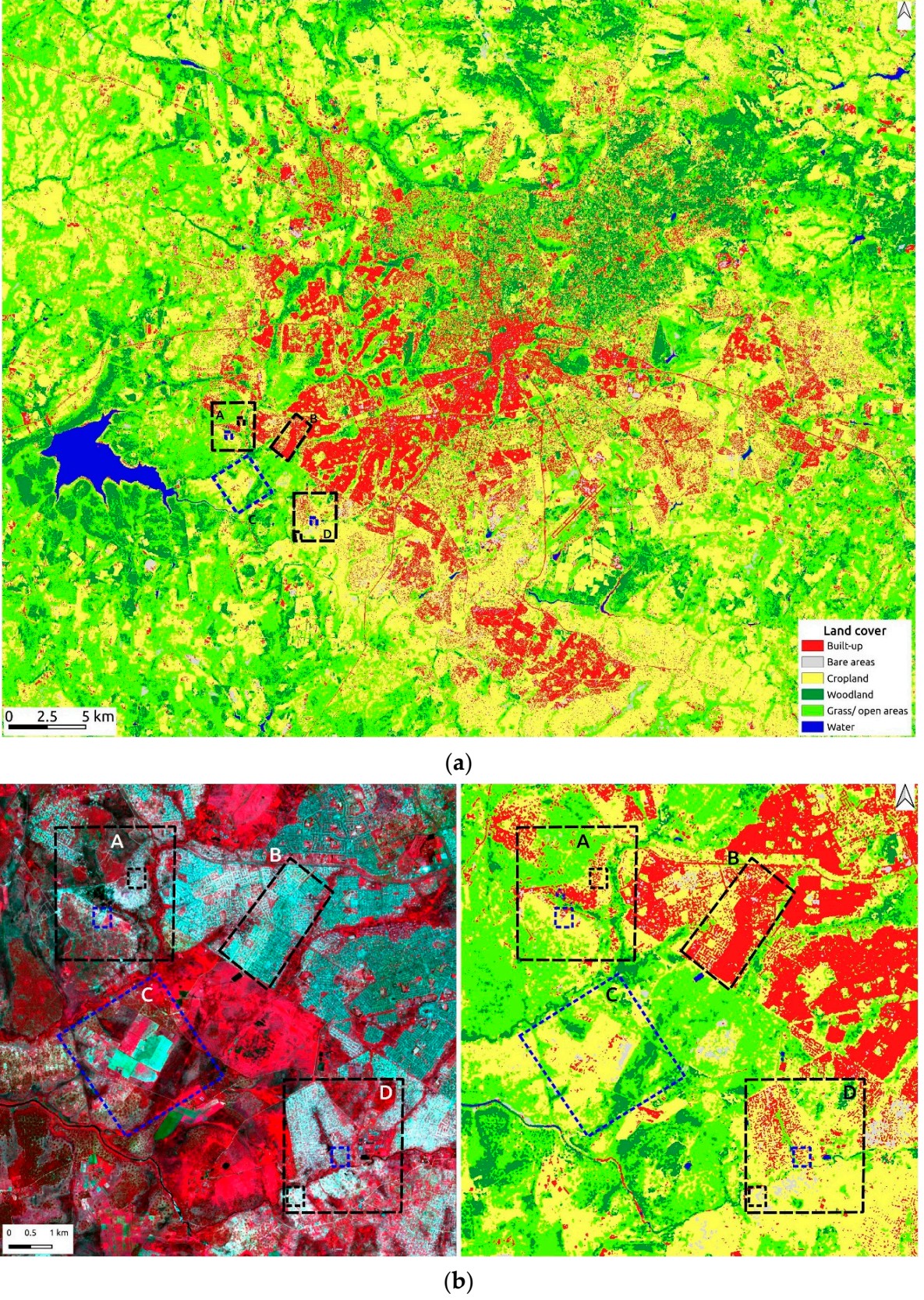

(**a**)

(**b**)

**Figure 7.** *Cont.*

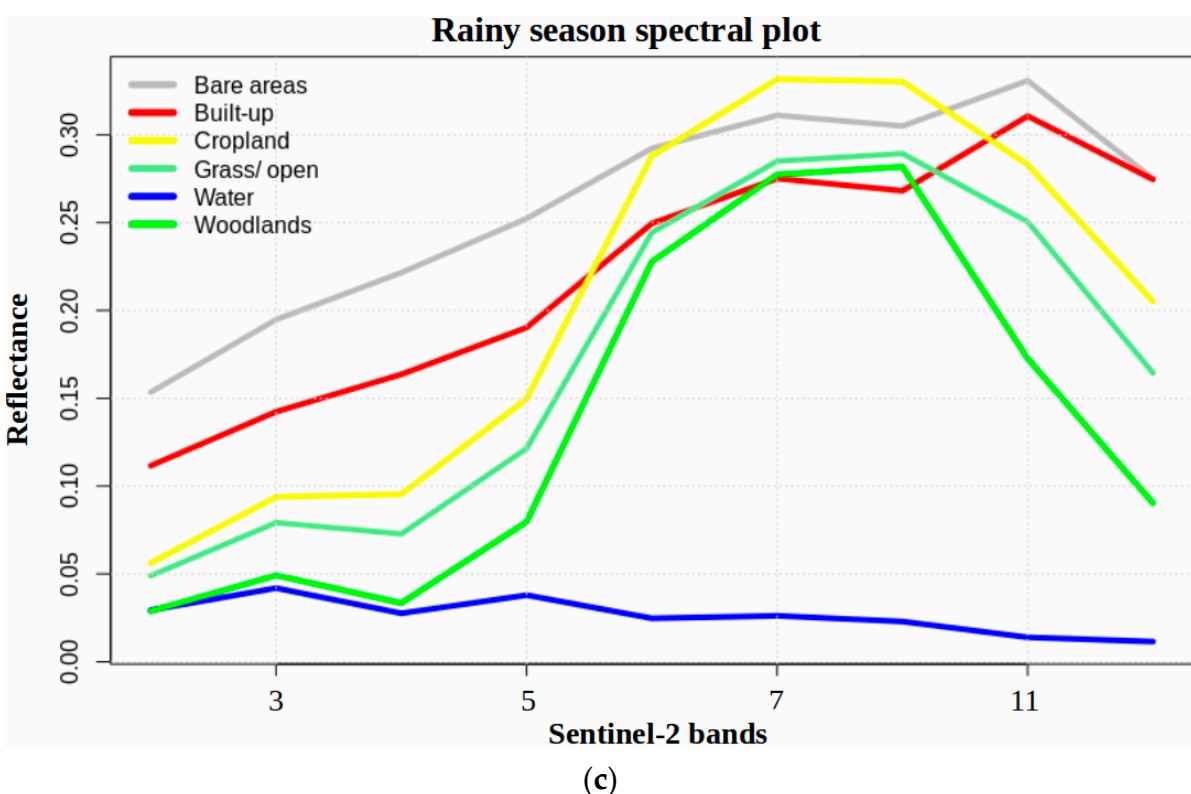

**Figure 7. (a)** S2RS land cover map: (A) Developing peri-urban; (B) developed urban area; (C) cropland area; and (D) developing peri-urban area. **(b)** Rainy season S2 (S2RS) imagery in false color (left), and S2RS land cover map (right). **(c)** Rainy season spectral plot for Harare.

The post-rainy season (S2PS) land cover map also shows classification errors (Figure 8a). For example, built-up patches are observed in the south-western part of the study area, which is mainly cropland. In the post-rainy season S2 (S2PS) imagery, cropland areas which are being prepared for an irrigation respond as bare soil and hence, it is difficult to separate them from built-up areas (location C in Figure 8b). The post-rainy season spectral profile indicates a narrow separability in the blue, green, and red bands for built-up and cropland areas, and a close spectral similarity in the other bands (Figure 8c). As was observed in the RSS2 land cover map (Figure 7a), built-up areas are also underestimated due to the fact that most of the houses in the low density residential areas are partially obscured by trees. Furthermore, grass/open areas are not correctly classified since these classes appear spectrally similar to cropland areas during the post-rainy season (Figure 8c). Indeed, grass/open areas have a low producer's accuracy of 37.8% (Table 3b), which show that the RF classifier missed most of the grass/open areas. This due to the fact that most crops are harvested during the post-rainy season. As a result, cropland areas are left with a crop residue or are covered by grass, which have the same spectral reflectance as grass/open areas (Figure 8c).

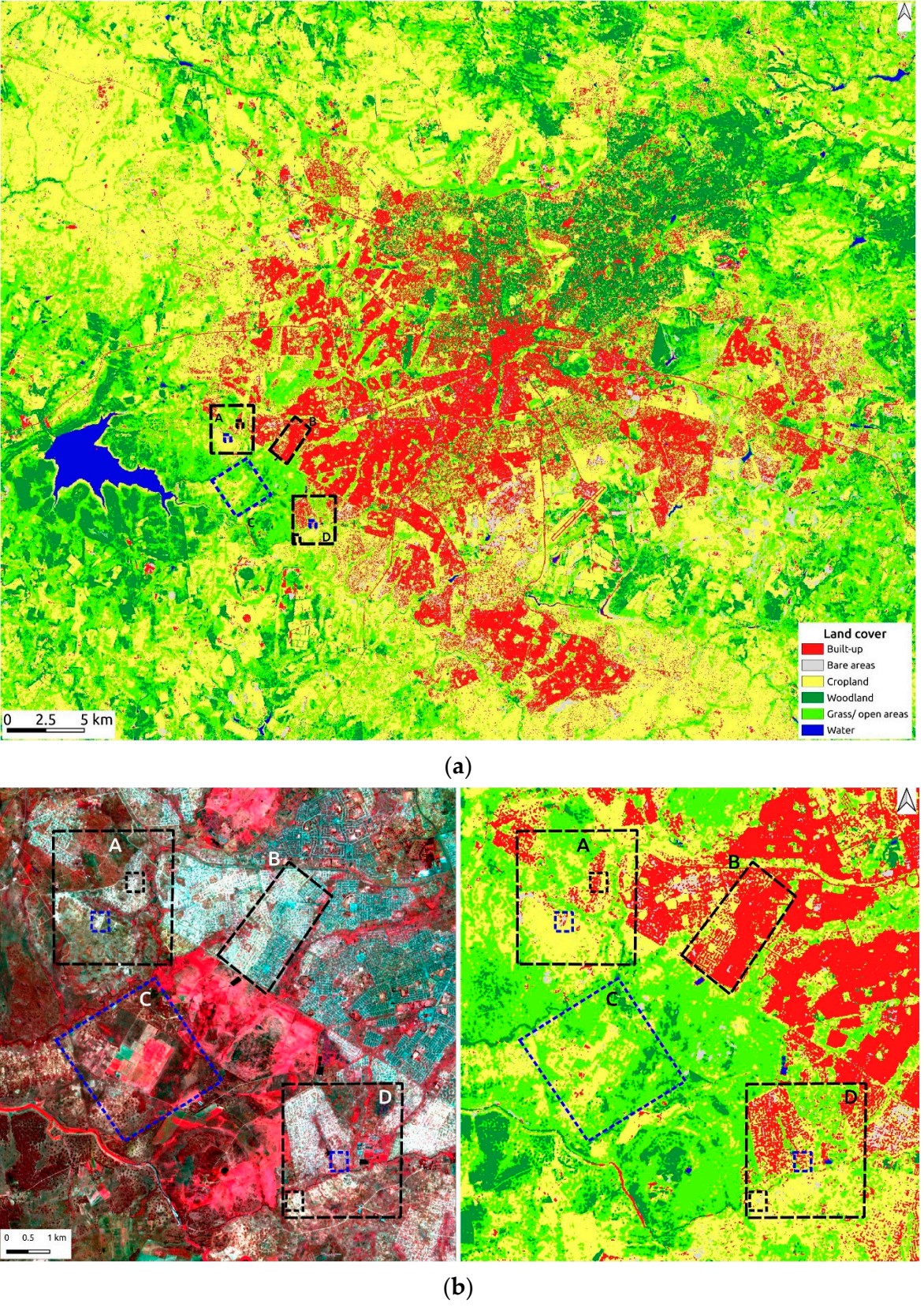

(**a**)

(**b**)

**Figure 8.** *Cont.*

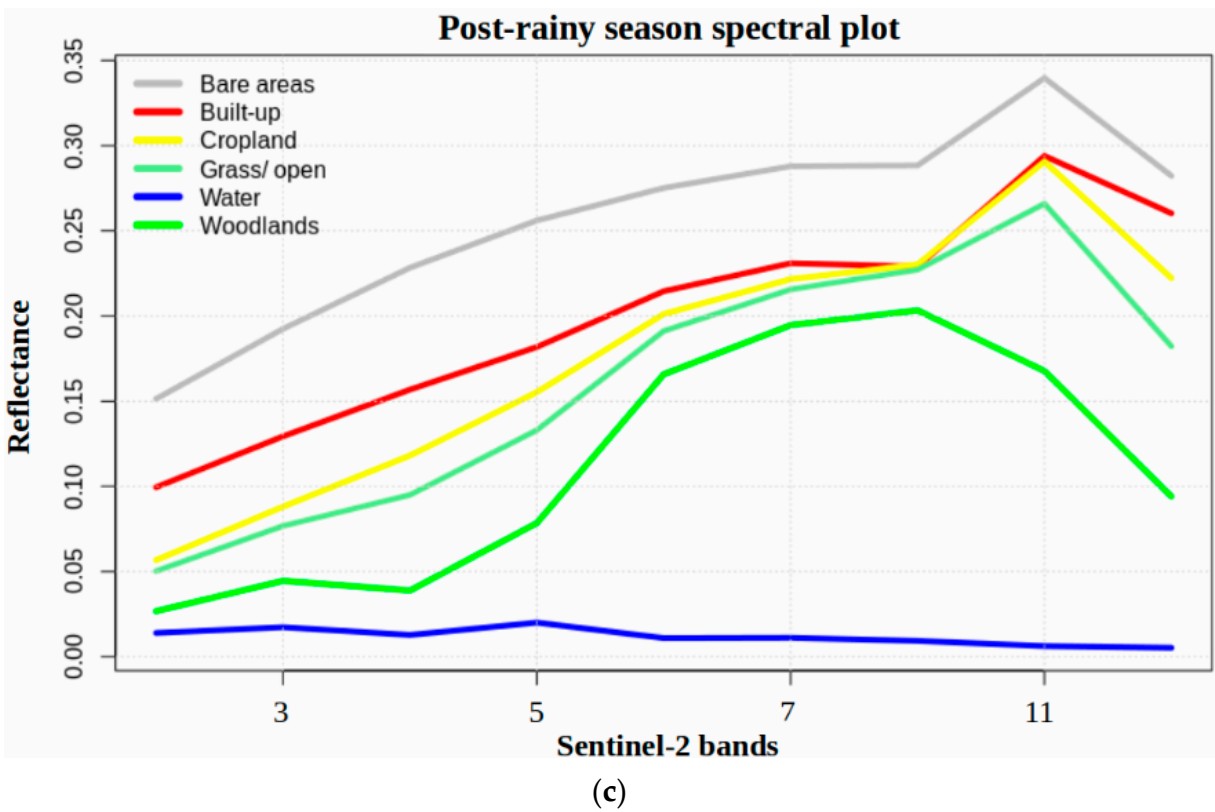

**Figure 8.** (**a**) S2PS land cover map: (A) Developing peri-urban; (B) developed urban area; (C) cropland area; and (D) developing peri-urban area. (**b**) Post-rainy season S2 (S2PS) imagery in false color (left), and S2PS land cover map (right). (**c**) Post-rainy season spectral plot for Harare.

The dry season (S2DS) land cover map also shows serious classification errors (Figure 9a). There is also an increased overestimation of built-up areas and spectral confusion between built-up and cropland areas (Figure 9b). This is due to the fact that most cropland are bare during the dry season. Figure 9c shows that the spectral reflectance between the built-up and cropland areas is quite close and therefore, it is difficult to separate these classes. However, there is an improvement in the classification of built-up class in low density residential areas since trees are leaf-off and hence, more built-up areas can be detected. As was observed in the S2RS and S2PS land cover maps, grass/open areas are also not correctly classified since these classes appear spectrally similar to cropland during the dry seasons (Figure 9c).

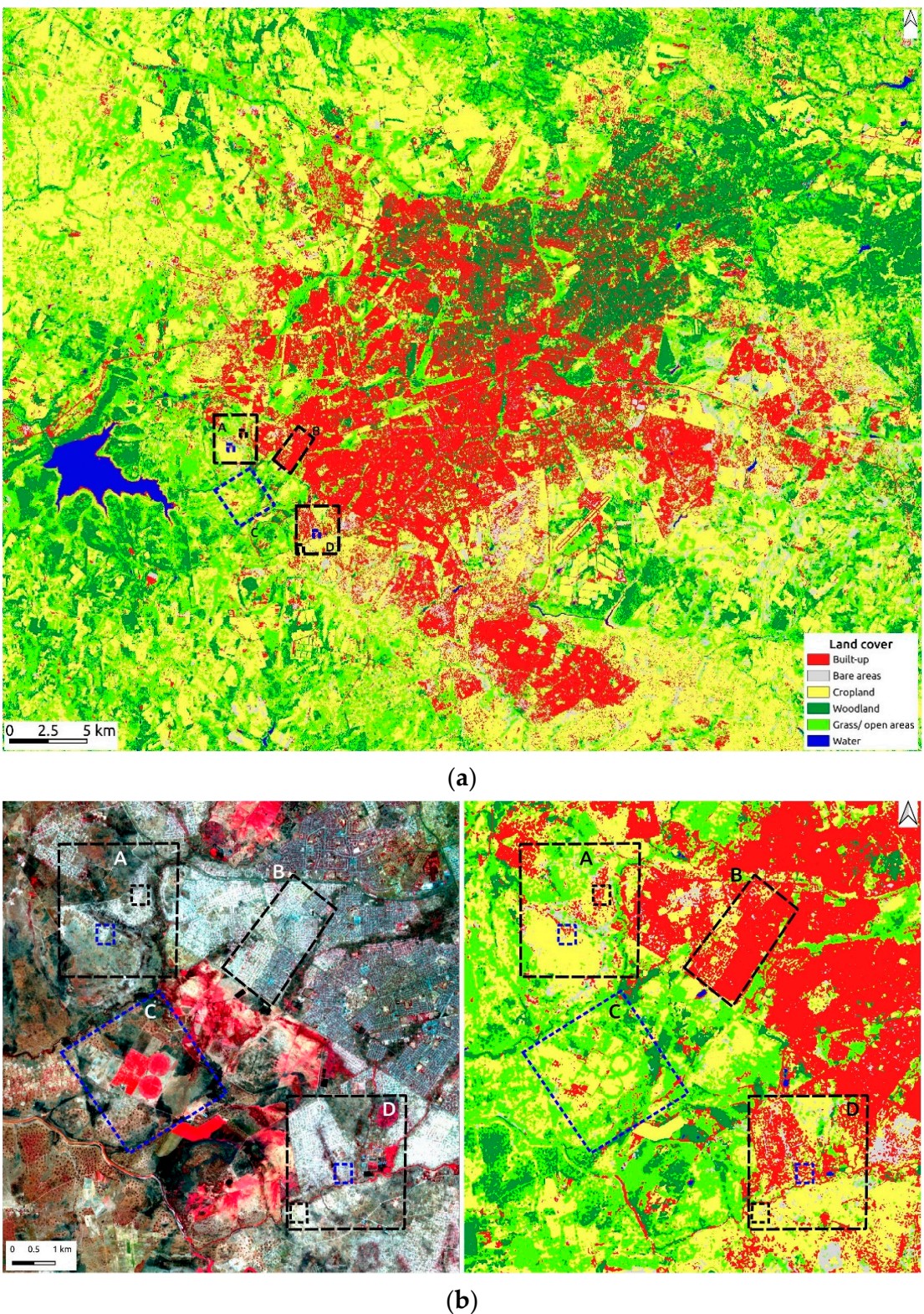

(**a**)

(**b**)

**Figure 9.** *Cont.*

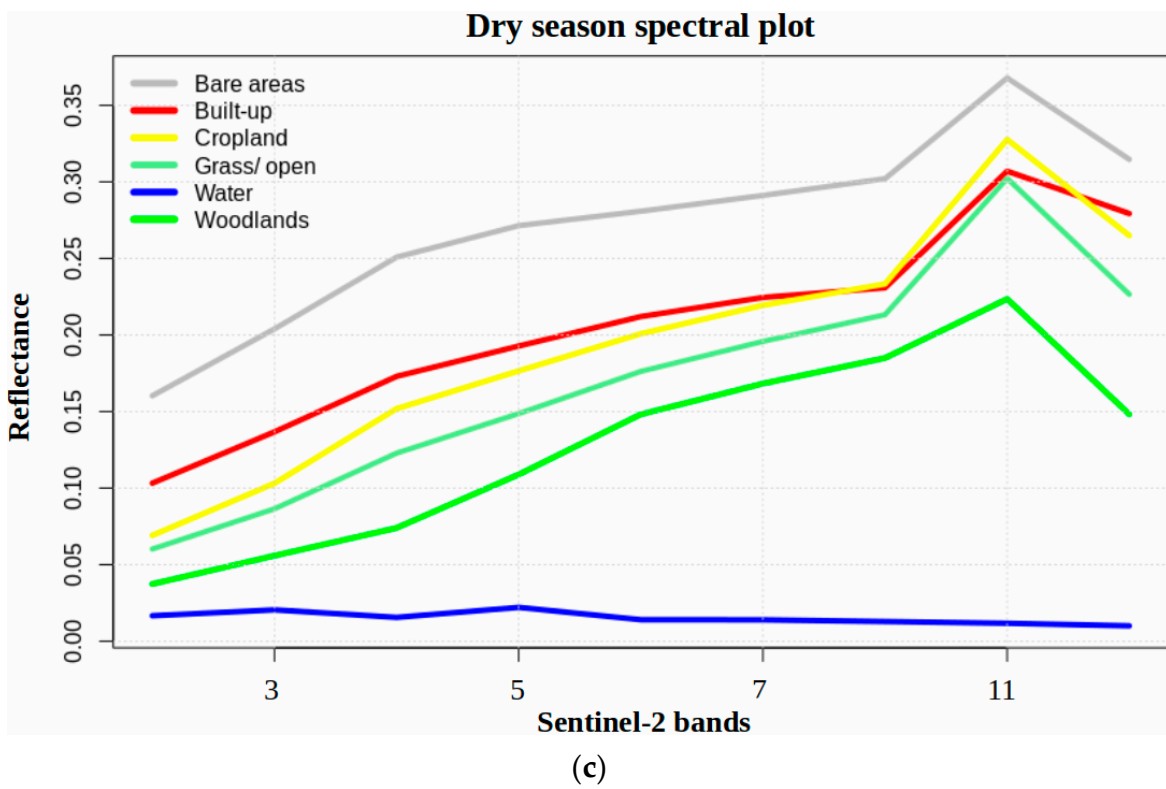

**Figure 9.** (**a**) S2DS land cover map: (A) Developing peri-urban; (B) developed urban area; (C) cropland area; and (D) developing peri-urban area. (**b**) Dry season S2 (S2DS) imagery in false color (left), and S2DS land cover map (right). (**c**) Dry season spectral plot for Harare.

The visual inspection of the multi-seasonal S2 (SS2) land cover map (Figure 10a) shows an improved classification compared to the SS1, S2RS, S2PS, and S2DS land cover maps. The SS2 land cover mapping results confirm the value of multi-seasonal information for improving land cover mapping. For example, the rainy season and post-rainy season S2 imagery is needed to separate crop fields from urban areas with significant amounts of asphalt and other impervious surfaces which are spectrally similar to bare soil in the dry season S2 imagery. However, there are cases where cropland or bare areas are still misclassified as built-up areas (Figure 10b). This is problematic particularly in newly developing peri-urban areas, and in areas where land is being prepared for cultivation (Figure 10b). Therefore, the SS2 land cover map results imply that the utility of multi-seasonal spectral information is limited, especially in peri-urban areas where cropland or grass/open areas are bare during the year. Note that bare cropland areas have a high soil to vegetation cover ratio, which appear spectrally similar to built-up areas, especially in newly developed peri-urban areas (Figure 3d–f).

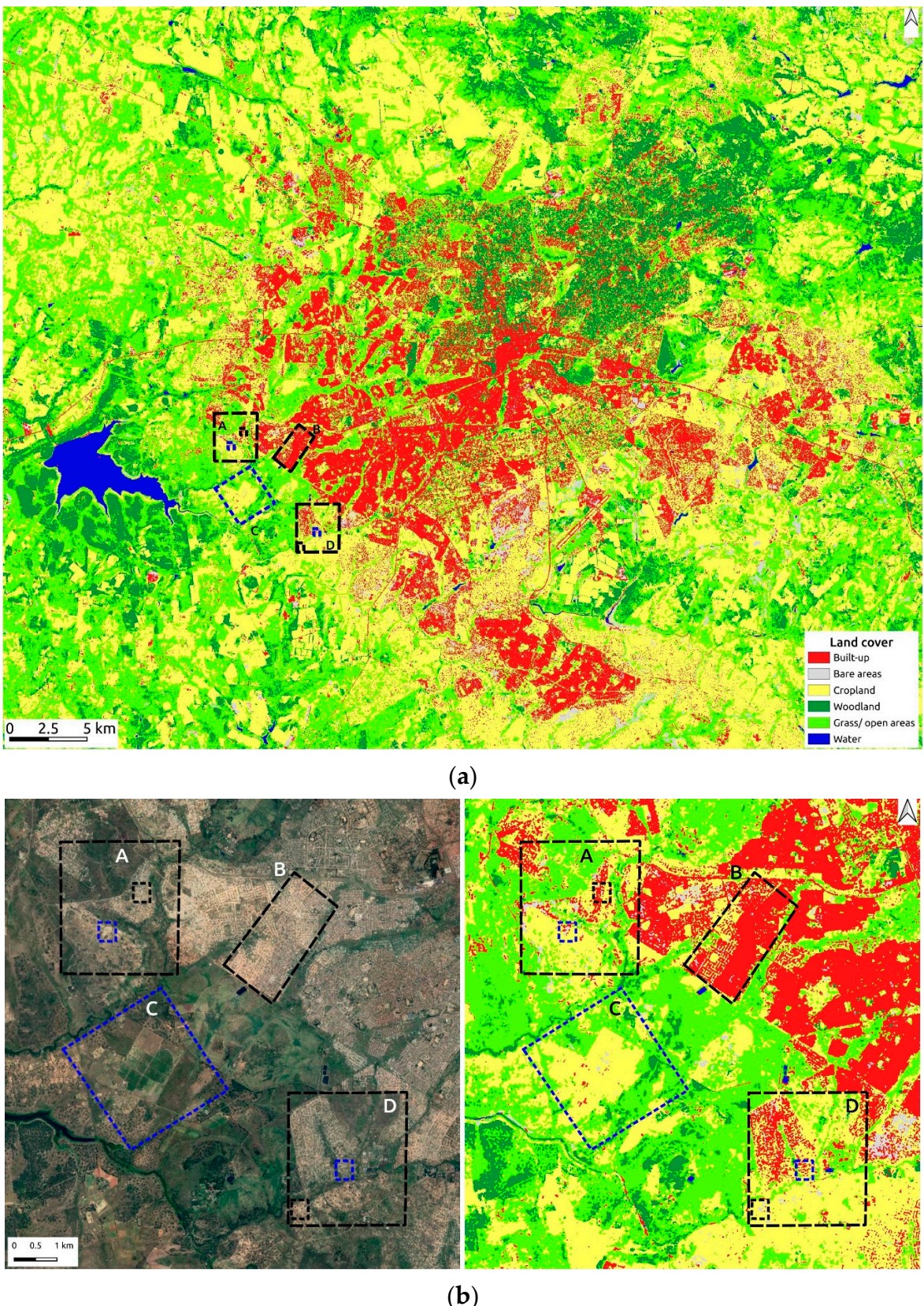

**Figure 10.** (**a**) SS2 land cover map: (A) Developing peri-urban; (B) developed urban area; (C) cropland area; and (D) developing peri-urban area. (**b**) Google Earth satellite (left), and (ii) SS2 land cover map (right).

Substantial improvement in land cover mapping is observed in the SC land cover map (Figure 11a). For example, the confusion between built-up areas, and cropland and bare/open areas is significantly reduced (Figure 11b). This is largely attributed to the contribution of S1 imagery. The analysis of the feature importance results show that the dry season S1 VH imagery is one of the most important features that contributes significantly to land cover mapping (Figure 11c). In the dry season S1 imagery, fallow cropland areas respond as bare soil as shown by the blue color in Figure 2c. These cropland areas are distinguishable from built-up areas in peri-urban areas. It is noteworthy that while cropland areas have varying backscatter (Figure 6c), depending on soil moisture, roughness, and crop canopy, they are recognizable from their parcel structure. Therefore, dry season S1 VH polarization, and rainy and post-rainy season S2 imagery are effective in separating cropland from built-up areas (Figure 11c).

### 4.1.3. Classification Accuracy Assessment

We conducted a rigorous (unbiased) accuracy assessment [58,59] for the Harare SC land cover map since there is more reference data for the study area. Note that the rigorous (unbiased) accuracy assessment is based on accuracy, which is calculated using area proportions not sample counts [58,59]. This means that absolute counts of the sample are converted into estimated area proportions using the equation provided by Olofsson et al. [59]. Therefore, the rigorous (unbiased) accuracy assessment results are reported in actual area units (km$^2$ or hectares) and area proportions, which are more meaningful than mere pixel counts. In this study, independent reference data were used for the accuracy assessment. The pixel was used as the spatial assessment unit since the SC land cover map was produced using a pixel-based RF approach. Originally, we determined a sample size of 676 points based on a stratified random sampling method [60]. However, land cover classes such as water and bare areas had less than 50 samples, while built-up and woodland classes had less than 100 samples. Therefore, we increased the built-up sample points to 200 (since it is one of the most important land cover classes), and bare areas and water to 50 sample points. In total, 876 sample points were used for the accuracy assessment. These sample points were interpreted from aerial photographs and very high resolution imagery from Google Earth™. We derived the area proportions and their confidence intervals, user's accuracy, and the producer's accuracy from the error matrix [59]. Note that the error matrix incorporates the standard error based on the total area of each land cover class. Therefore, the land cover class area estimates are not biased.

Table 4 shows the mapped land cover class area estimates, ±95% confidence interval (CI), the user's accuracy, and the producer's accuracy. The accuracy assessment results show that the overall accuracy is 72.5%, while the individual class accuracies vary significantly (Table 4). The user's accuracy of estimated area proportion is higher than the producer´s accuracy of estimated area proportion for the built-up class. Therefore, there are more errors of omission given that some of the built-up areas were missed, especially in vegetated low density areas. The bare areas class has low individual accuracies, which indicates severe classification problems. This is attributed to the high class error for the bare areas class (Figure 5). Generally, there are small or no marginal differences in individual accuracies for cropland, woodland, and grass/open areas classes. However, there are more errors of commission for the cropland class, which suggests an overestimation of this class. For the water class, there are more errors of omission, which means that the RF classifier missed most of the water areas.

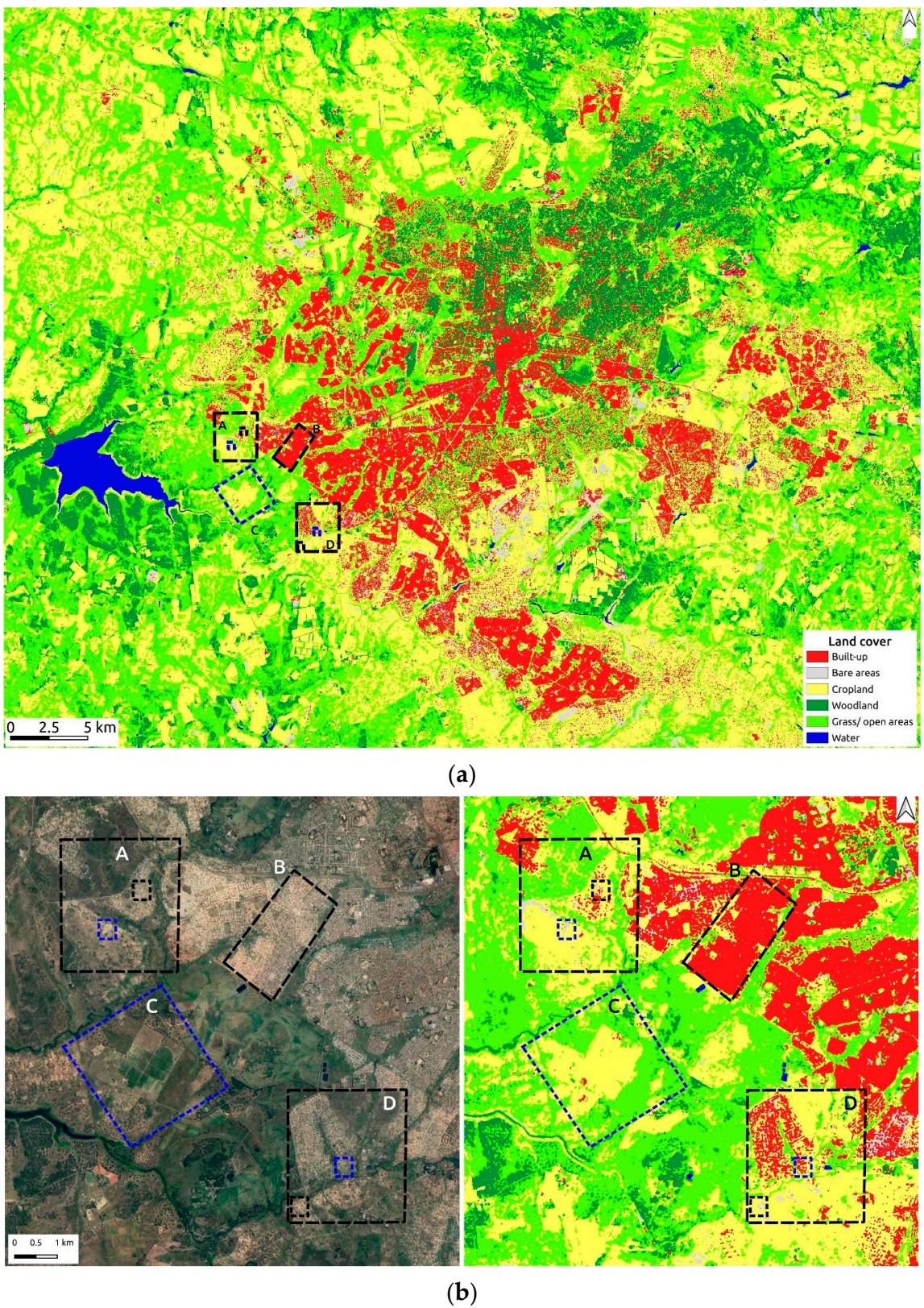

(**a**)

(**b**)

**Figure 11.** *Cont.*

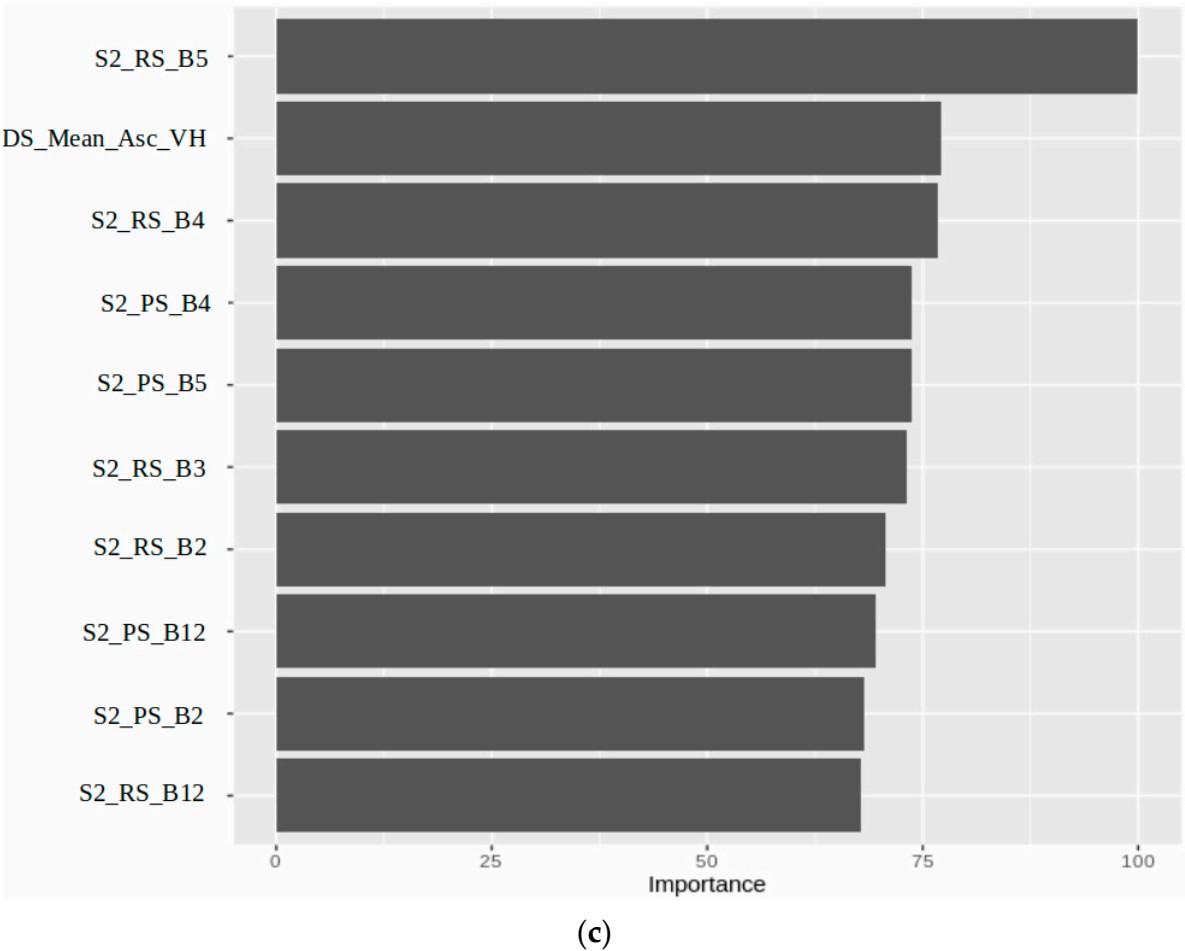

**(c)**

**Figure 11.** (**a**) SC land cover map: (A) Developing peri-urban; (B) developed urban area; (C) cropland area; and (D) developing peri-urban area. (**b**) Google Earth satellite imagery (left), and (ii) SC land cover map (right). (**c**) Random forest (RF) feature importance for the SC land cover map.

**Table 4.** Summary of the rigorous (unbiased) accuracy assessment.

| Class | Area (km²) | ±95% CI (km²) | User's Accuracy (%) | Producer's Accuracy (%) |
|---|---|---|---|---|
| Built-up | 399.2 | 20.8 | 91.5 | 72.8 |
| Bare areas | 23.4 | 8.3 | 12 | 39.1 |
| Cropland | 929.5 | 38.3 | 74.3 | 77.8 |
| Woodlands | 314.4 | 45.1 | 68.6 | 68.9 |
| Grass/open areas | 1,149.8 | 25.8 | 70.2 | 70.2 |
| Water | 37.1 | 7.3 | 98 | 65.8 |
| Total | 2853.8 | | | |

### 4.2. Land Cover Mapping in Other Major Urban Centers in Zimbabwe

#### 4.2.1. Comparison of Model Overall Accuracy

We also evaluated the performance of the RF classifier for the other major urban centers in Zimbabwe. In terms of the overall accuracy, the test models results vary for all urban centers (Figure 12). For Bulawayo, the SC test model has the highest overall accuracy followed by the SS2 test model. The S2PS and S2DS test models have a relatively higher overall accuracy than the SS1 and S2RS test models for Bulawayo. This suggests that SS1 and S2RS data have severe limitations for land cover mapping in the study area. Figure 13a shows classification problems, which are conspicuous in the land cover maps. For example, some bare areas in the far western part of the city (which is generally rugged terrain) are misclassified as built-up areas, while built-up areas in core urban settlements are completely omitted or misclassified as woodland areas in the SS1 land cover map (Figure 13ai). The visual analysis also reveals significant classification problems in the S2RS, S2PS, S2DS, and SS2 land cover maps (Figure 13aii–v). The S2RS land cover map shows that bare areas are overestimated in the north-western part of the city (Figure 13bii), while some grass/open areas are misclassified as water areas. In contrast, the S2PS, S2DS, and SS2 land cover maps show that grass/open areas are misclassified as built-up areas in the north-western part of the study area (Figure 13biii–v). The classification problems are likely attributed to the spectral similarity between the grass/open areas (which are bare during the post-rainy and dry season) and built-up areas. However, a substantial improvement in the classification accuracy is observed in the SC land cover map (Figure 13avi,bvi). The overall accuracy for the SC land cover map is 7.3% higher than the SS2 land cover map.

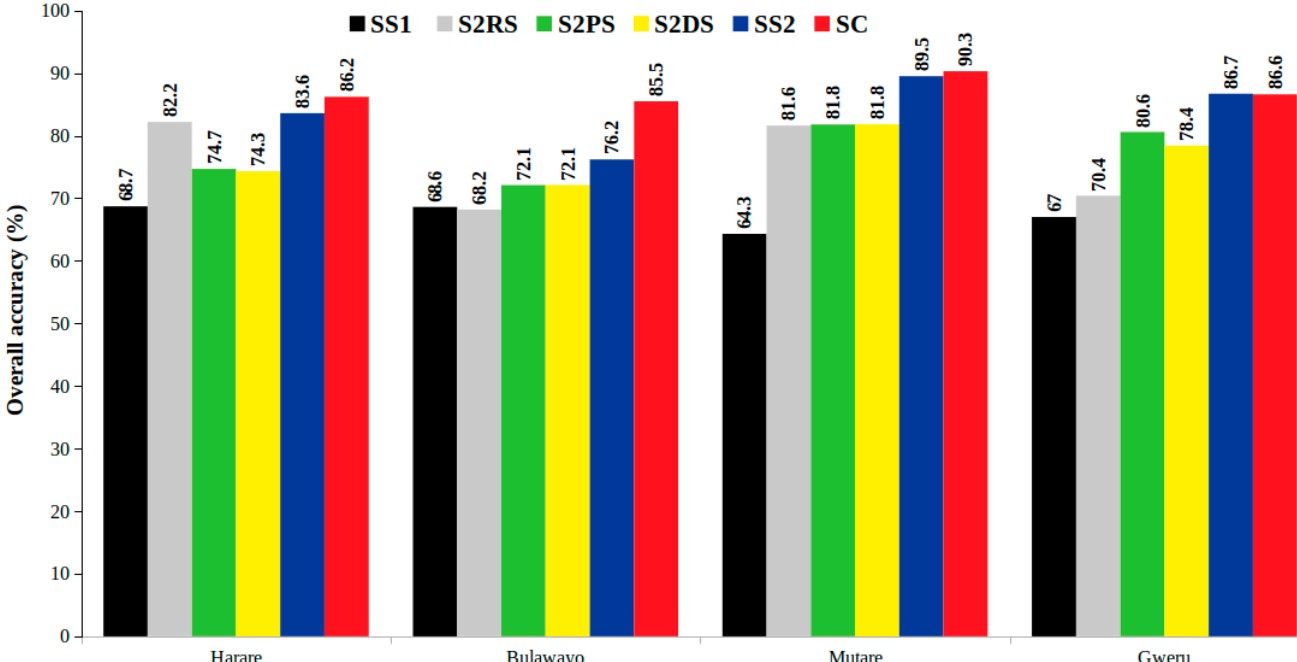

**Figure 12.** The overall accuracy for the test models in each city.

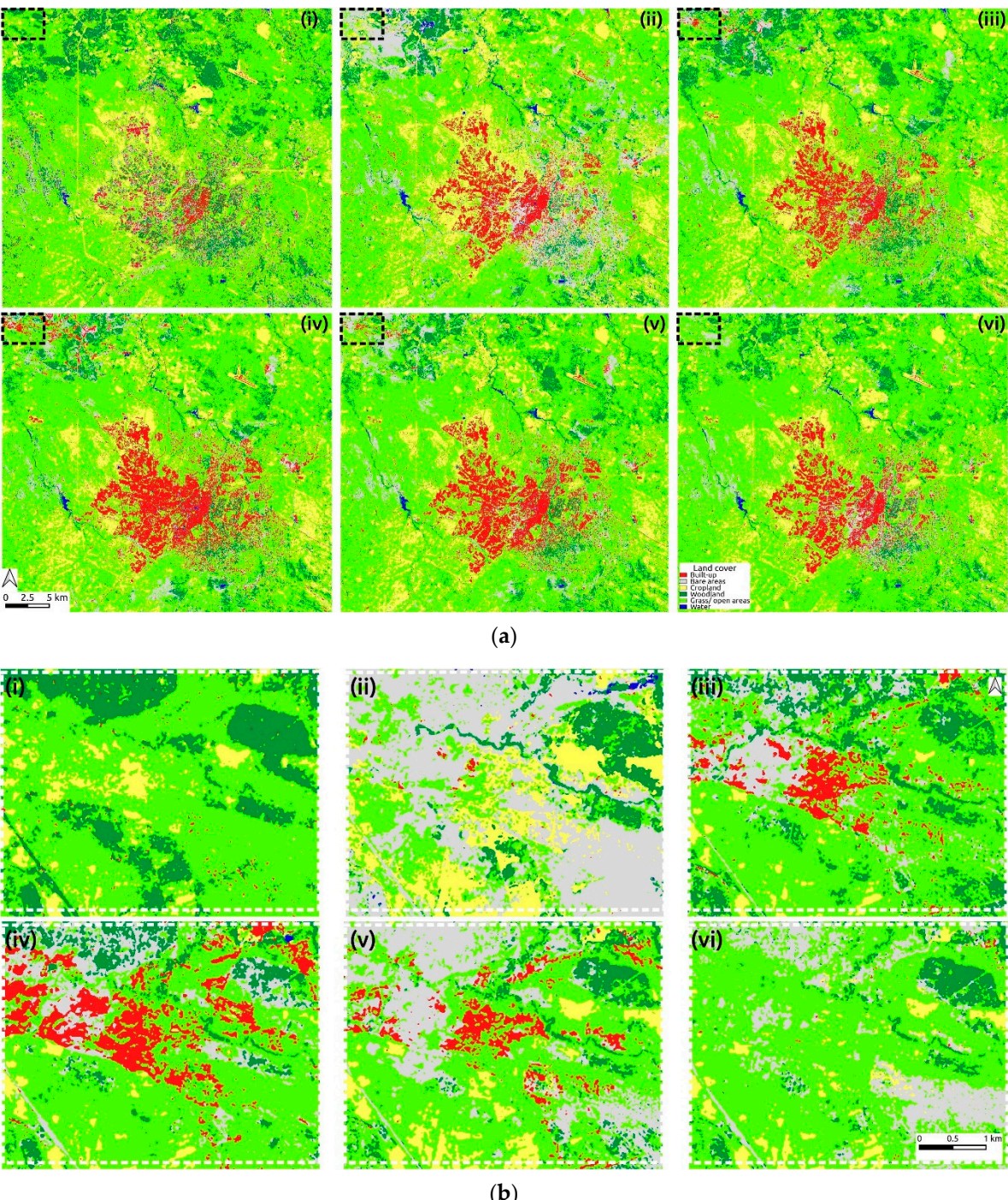

**Figure 13.** (*a*) (i) SS1; (ii) S2RS; (iii) S2PS; (iv) S2DS; (v) SS2; and (vi) SC land cover maps for Bulawayo. (**b**) Bulawayo: (i) SS1; (ii) S2RS; (iii) S2PS; (iv) S2DS; (v) SS2; and (vi) SC land cover map subsets.

For Mutare, the SC and SS2 test models have the highest overall accuracy, while the SS1 test model has the lowest overall accuracy (Figure 12). The S2RS, S2PS, and S2DS test models achieve relatively moderate overall accuracies (Figure 12). The visual analysis shows a substantial underestimation of the built-up areas in the S1 land cover map (Figure 14a), which clearly indicates the limitations of using only S1 data in Mutare. Interestingly, the built-up areas with a high backscatter, which are oriented orthogonal to the S1 sensor look direction, are misclassified as bare areas. In addition, the built-up areas in developed urban areas which are not oriented towards the S1 sensor look direction are misclassified as woodland areas, especially in high density areas located in the southern part of the city (Figure 14a). In contrast, a substantial overestimation of built-up areas is observed in the S2RS, S2PRS, and S2DS land cover maps (Figure 14b–d). This suggests that mono-seasonal S2 data (which are S2RS, S2PS, and S2DS data) are not optimal for land cover mapping in the study area. Although a high classification accuracy is observed in both SS2 and SC land cover maps, the visual analysis shows an improved mapping of built-up areas in the latter (Figure 14e,f).

Similarly, the SC and SS2 test models have the highest overall accuracy for Gweru (Figure 12). However, there are marginal differences between the SS1 and S2RS test models, and between the S2PS and S2DS test models. This also suggests that mono-seasonal data are not adequate for land cover mapping for Gweru. Figure 15 shows an underestimation of the built-up areas in the S1 land cover map and a substantial overestimation of built-up areas for the S2RS, S2PRS, and S2DS land cover maps. In particular, severe classification problems are observed in the S2RS land cover map (Figure 15b). For example, some grass/open areas in the south and south-western part of the city are misclassified as built-up areas. This is likely attributed to the low rainfall received in that study area during this period. As a result, grass/open and cropland areas respond as bare soil and hence, it is difficult to separate them from built-up areas. It is important to note that the SS2 land cover map has a slightly higher overall accuracy than the SC land cover map for Gweru. However, the visual analysis shows that the SC land cover map is better than the SS2 land cover map since the latter has some misclassified built-up areas in the southern part of the study area.

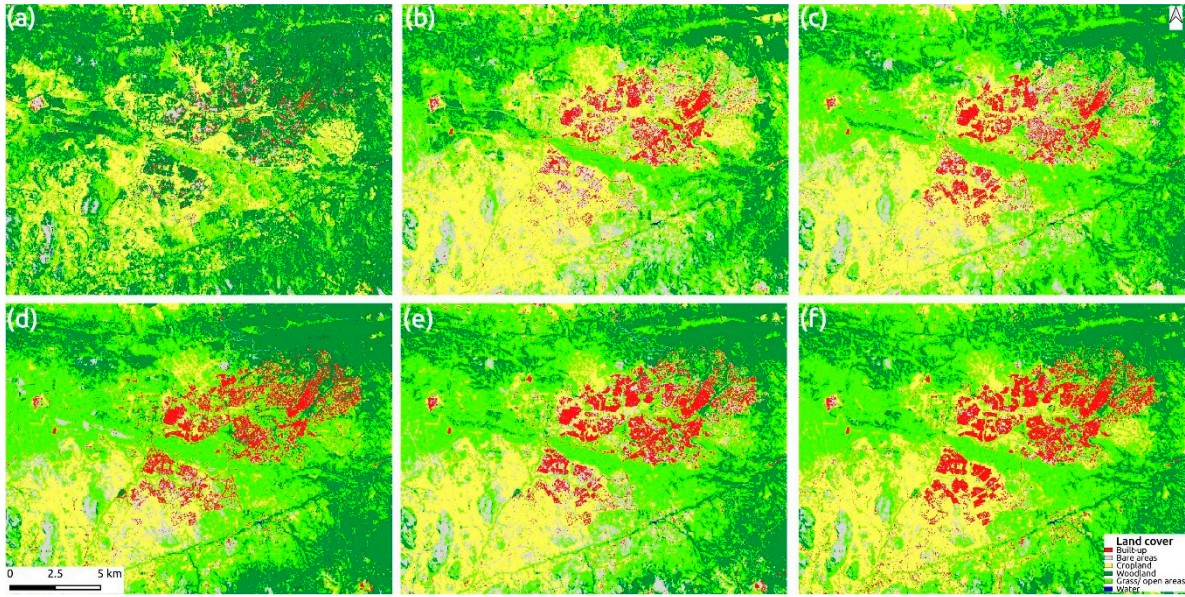

**Figure 14.** (**a**) SS1; (**b**) S2RS; (**c**) S2PS; (**d**) S2D; (**e**) SS2; and (**f**) SC land cover maps for Mutare.

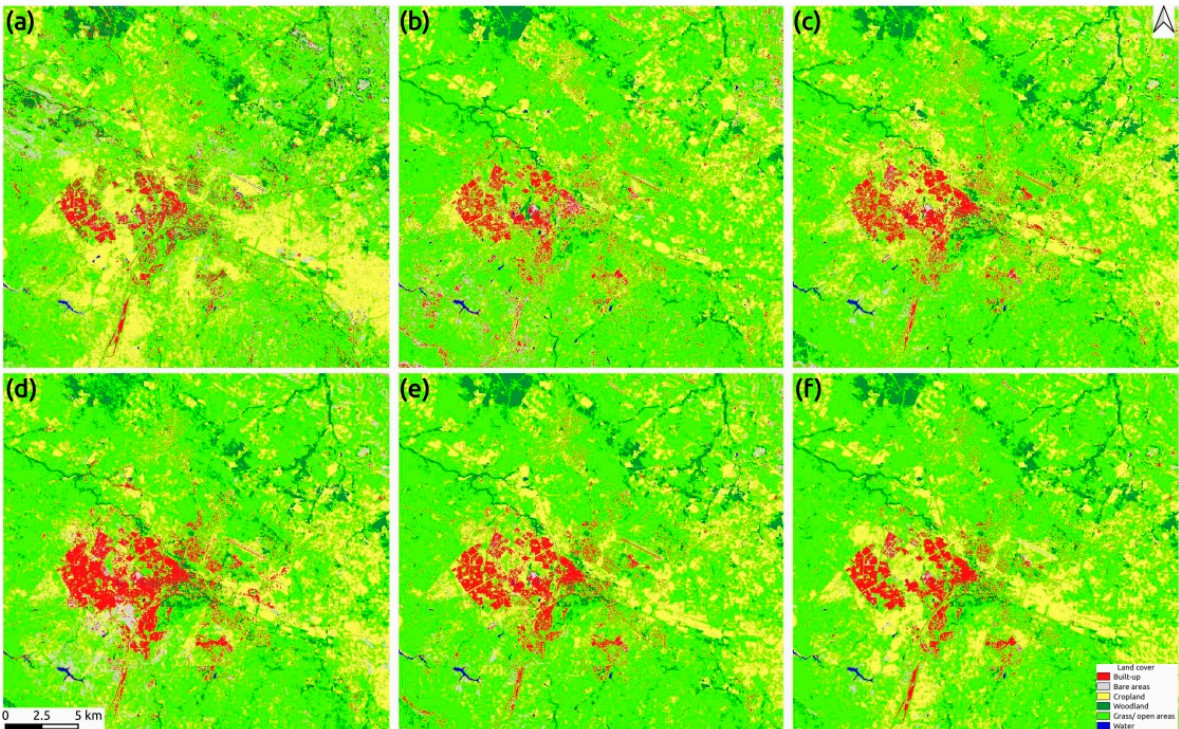

**Figure 15.** (**a**) SS1; (**b**) S2RS; (**c**) S2PS; (**d**) S2DS; (**e**) SS2; and (**f**) SC land cover maps for Gweru.

### 4.2.2. Random Forest Feature Importance

We also evaluated the performance of the SC test model based on feature importance. Figure 16 shows the relative importance of the contribution of the top 10 features out of the 39 multi-seasonal S1 and S2 bands used for land cover mapping. The results show that rainy and post-rainy season S2 bands, and dry and rainy season S1 VV and VH features are important for land cover mapping in all major urban areas. In general, band 2 (Blue), band 3 (Green), band 4 (Red), and band 5 (Vegetation red edge) from the rainy and post-rainy seasons are the most important features for Harare, Bulawayo, and Mutare. This is probably due to the fact that during the rainy and post-rainy seasons, the vegetation and built-up areas show the greatest difference (Figures 7c and 8c). In addition, the importance of dry season VV and VH polarization features shows that the complementary information derived from S1 data improves land cover mapping. This is due to the fact that the SAR sensor is effective at capturing the structure and dielectric properties of the Earth surface materials [61]. Note that the dry season VV and VH, and rainy season VH polarization are selected as important features for Bulawayo (Figure 16b). This is mainly attributed to the semi-arid landscape and dominant land cover type in the study area. Generally, Bulawayo receives a low rainfall and is primarily dominated by grass/open areas.

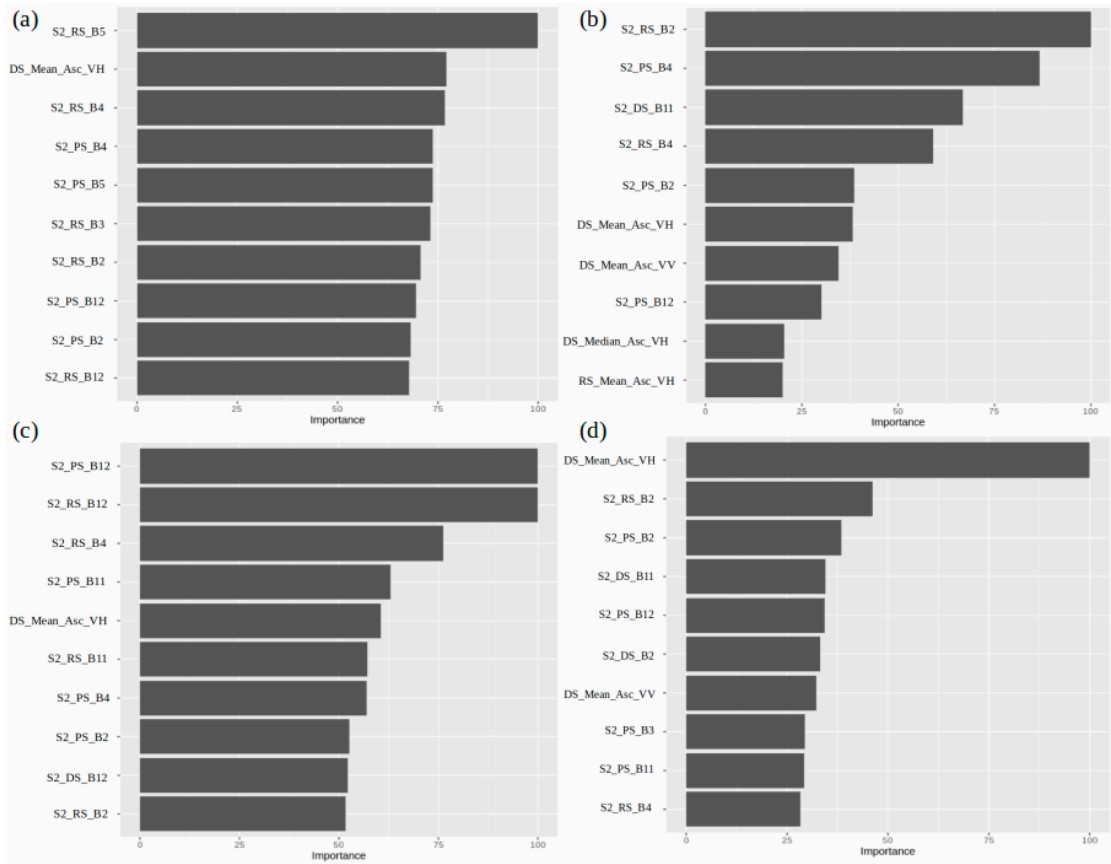

**Figure 16.** RF feature importance measures for the SC land cover map: (**a**) Harare; (**b**) Bulawayo; (**c**) Mutare; and (**d**) Gweru.

## 5. Discussion

The recent drive to provide current and accurate geospatial information for the UN SDGs and UN New Urban Agenda (NUA) require reliable approaches that improve land cover mapping in urban and peri-urban areas in Sub-Sahara Africa. The availability of free and high-quality optical and SAR data from the Copernicus programme under the European Space Agency offer opportunities to improve land cover mapping. We evaluated the utility of multi-seasonal S1 (SS1), mono-seasonal S2 (S2RS, S2PS, and S2DS), multi-seasonal S2 (SS2), and multi-seasonal composite S1 and S2 (SC) data for mapping land cover in four major urban centers in Zimbabwe. Our results indicate that the SC data produced superior discrimination of built-up areas and other land cover classes relative to SS1, mono-seasonal S2, and SS2 data in the study area. The overall accuracy scores for the SC land cover maps are above 85% for all urban centers. In particular, the SC data improved the overall accuracy to 2.6%, 9.3%, and 0.8% for Harare, Bulawayo, and Mutare, respectively compared to only the SS2 data. Although the overall accuracy is slightly higher for the SS2 data for Gweru, the visual analysis reveals that the SC data significantly improves land cover mapping for all urban centers.

It is noteworthy that the observed variability in S2RS, S2PS, S2DS, and SS1 training and testing models (Figure 5) highlight limitations of using only mono-seasonal S2 and SS1 data for mapping land cover in the study area. For example, an increase in the overall accuracy of 4%, 17.3%, 8.7%, and 16.2% is achieved for Harare, Bulawayo, Mutare, and Gweru when using SC data relative to only the rainy season S2 data. Our results indicate that SS1 data is not suitable for land cover mapping in the study area. An increase in the overall accuracy that varies between 17% and 26% is achieved when SC data are used for land cover mapping instead of only SS1 data for all urban centers. We observed that settlement areas which are not oriented towards the SAR sensor look direction have a significant cross-

polarized component. This leads to a severe confusion between built-up and woodland areas [62] and hence, poor accuracy in the SS1 land cover map (Figure 6a). Previous studies have observed this scattering ambiguity and confusion since the urban environments are comprised of various natural and man-made targets, different orientations, various shapes and sizes [54,63,64]. Furthermore, the scattering behavior also depends on the geometry of terrain elements and their electromagnetic characteristics [65]. Therefore, interpreters of S1 data must be aware of these scattering effects which tend to complicate the backscatter analysis in urban and peri-urban areas [66]. Although leaf-on S2 data from the rainy and post-rainy season are assumed to improve the separation of built-up areas from cropland and bare areas, land cover mapping results generally varied in this study areas. For example, rainy season S2 data produced relatively higher levels of overall accuracy for Harare and Mutare compared to Bulawayo and Gweru. This suggests that local climatic variability significantly impacts land cover mapping accuracy.

This study revealed the importance of seasonality and the combination of S1 and S2 data, especially when attempting to map land cover in fast developing peri-urban areas. Clearly, SS1 and mono-seasonal S2 data are not effective for mapping land cover in all urban centers given the inter- and intra-annual variability in land cover types (Figure 3). Furthermore, our results also indicate the limitations of using only multi-seasonal S2 data. Although previous studies [43] demonstrated the use of multi-seasonal optical imagery for separating different land cover classes, the spectral confusion between built-up areas and cropland and bare areas still occur, especially in peri-urban areas. For example, fallow or post-harvest cropland, bare areas, and newly developing peri-urban areas are confused with one another during the year (Figure 3d–f). In addition, grass/open areas are also easily confused with cropland if assessed during one season. However, the use of combined SC data brings significant benefits. The feature importance analysis results indicate that rainy and post-rainy season S2 bands, and dry season S1 VV and VH polarization improve land cover mapping in urban and peri-urban areas. Although we observed different scattering effects and confusion between built-up areas and woodland areas in SS1 data, the cropland areas were not confused with built-up areas in all urban centers. This is likely attributed to the effectiveness of VV and VH polarization in separating cropland from built-up areas in the study area. In general, backscatter from cropland areas is usually composed of surface scattering from the soil, volume scattering from the plants, and a soil-vegetation interaction component. In the study area, surface scattering from the soil is dominant in the early rainy (growing) season, while volume scattering dominates during the peak growth period (mid-rainy season and early post-rainy season) (Figure 2a–c). During the harvesting and post-harvesting periods, a mixture of surface and volume scattering is dominant. While this makes information extraction difficult between bare areas and cropland in the study area, the scattering effects make it easier to separate built-up areas from cropland. This is very important since the spectral confusion between cropland and built-up areas causes severe misclassifications when only optical data are used for land cover mapping, especially in newly developing peri-urban areas [10]. Therefore, the combination of backscatter derived from S1 data [61], and spectral and seasonal information derived from S2 data substantially improved land cover mapping in the study area.

While the study demonstrated the capability of SC data for improving land cover mapping in the study area, some challenges still need to be addressed. First, the scattering ambiguity and confusion observed in S1 (VV and VH, ascending orbit) data still pose problems for SAR data interpretation and analysis. Therefore, there is a need to explore alternative SAR data analysis techniques in order to further improve land cover mapping accuracy. Second, the unavailability of HH and HV polarization in descending orbit during the study period limits the full analysis of S1 data, and understanding its impact on land cover mapping. It is important to consider a 2- or 3-year study period since there are higher chances of getting descending orbit HH and HV polarization data in the study area. Third, the lack of anniversary or near-anniversary very high resolution imagery or ground reference data for the other major urban centers (Bulawayo, Mutare, and Gweru) limits

the ability to conduct a rigorous accuracy assessment as was done for Harare. Although the use of very high resolution satellite imagery from Google Earth is a widely adopted method for accuracy assessment, the on-site field data collection and very high resolution imagery would contribute to more reliable reference data. In this regard, there is a need to collaborate with local city and government authorities, as well as universities in order to collect reference data using field surveys and low-cost unmanned aerial vehicles (UAV). Nevertheless, our land cover mapping approach shows a potential to map land cover in other medium or small urban centers, which are characterized by fragmented built-up developments in peri-urban areas.

## 6. Conclusions

In summary, several conclusions can be drawn from this study. First, this study has revealed that the use of S1 data is not effective for mapping land cover in urban and peri-urban areas in the study area. Second, this study also illustrated limitations of using mono-seasonal S2 and multi-seasonal S2 data for land cover mapping, especially in complex different urban landscapes in Zimbabwe. Third and more importantly, this study demonstrated that using multi-seasonal composite S1 and S2 (SC) data as input for classification results in accurate land cover maps. This is attributed to the sensitivity of S1 data to detect different target surfaces and the ability to separate cropland from built-up areas, as well as the capacity of multi-seasonal S2 data to identify phenological changes. Given the availability of high volumes of medium to high resolution satellite data (Landsat, S1, S2, etc.), there is an increased need for cost-effective approaches that can be used to improve land cover mapping, especially in fast-developing urban and peri-urban areas in Sub-Sahara Africa. Thus, the assessment of multi-seasonal S1 and S2 data and different random forest models to improve the land cover mapping is of critical importance. This study revealed that SC random forest training and test models performed more effectively in different urban landscapes. The combination of multi-seasonal S1 and S2 data improved the overall classification significantly, mostly improving the mapping of built-up areas in peri-urban and low density residential areas. Therefore, our land cover mapping approach contributes to land cover studies, which can be applied to other urban areas in Zimbabwe and in Sub-Sahara Africa.

**Supplementary Materials:** The following are available online at https://www.mdpi.com//1/1/9/s1. Figure S1: Sentinel imagery for Harare: (a) S1RS; (b) S1PS; (c) S1DS; (d) S2RS; (e) S2PS; and (f) S2DS. S1 imagery is displayed in false color RGB (VV, VH, VV), while S2 imagery is displayed in false color RGB (8,4,3); Figure S2: Newly-developing peri-urban area: (a) S1RS; (b) S1PS; (c) S1DS; (d) S2RS; (e) S2PS; (f) S2DS subsets, and Google Satellite imagery (locations A and B). The blue rectangle shows sparse built-up areas in newly developed peri-urban areas, while the black rectangle shows typical cropland areas; Figure S3: Sentinel imagery for Bulawayo: (i) S1RS; (ii) S1PS; (iii) S1DS; (iv) S2RS; (v) S2PS; and (vi) S2DS. S1 imagery is displayed in false color RGB (VV, VH, VV), while S2 imagery is displayed in false color RGB (8,4,3); Figure S4: Sentinel imagery for Mu-tare: (a) S1RS; (b) S1PS; (c) S1DS; (d) S2RS; (e) S2PS; and (f) S2DS; Figure S5: Sen-tinel imagery for Gweru: (a) S1RS; (b) S1PS; (c) S1DS; (d) S2RS; (e) S2PS; and (f) S2DS.

**Author Contributions:** C.K. was responsible for designing and conducting the study. He classified all the Sentinel-1 and Sentinel-2 data and prepared the manuscript; O.W.K. was responsible for preparing reference and validation data, as well as revising the manuscript; E.C. and J.G. were responsible for revising and editing the manuscript. All authors have read and agreed to the published version of the manuscript.

**Funding:** This research received no external funding.

**Institutional Review Board Statement:** Not applicable.

**Informed Consent Statement:** Not applicable.

**Data Availability Statement:** Not applicable.

**Conflicts of Interest:** The authors declare no conflict of interest.

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
