# Peer review of "Mapping Urban and Peri-Urban Land Cover in Zimbabwe: Challenges and Opportunities"

_2673-7418, doi:10.3390/geomatics1010009_

Round 1

Reviewer 1 Report

The authors assess the contribution of various inputs and transformations of S1/S2 data for a land cover application over a set of cities in Zimbabwe. The application is novel, and the authors clearly justify the rational behind it, aligning well with the UN SDG. The manuscript is generally well written, but it is tediously long. I believe some sections can be shortened and some material can be transferred to a supplementary appendix if the authors see them as critical.

For example, a lot of Sentinel information is perhaps unnecessary (there is no need to me at least, to present tables describing the sentinel bands, a proper citation should suffice).

Moreover, although you could present the training accuracy for Harare as it is a more detailed section, I find no need to present whole sections for OOB accuracy/error results for the rest of the cities (could go in the supplementary material). The validation (testing accuracy) related results, should suffice. It is important to understand that the OOB accuracy is considered a pseudo-test metric – as it is an out of sample validation and some researchers use it as an alternative to an independent test. Using both sequentially, and deducing different conclusions from each one is confusing for the readers.

Overall, I commend the authors for the work done, and I certainly believe their work is worth publishing, upon some changes refining the structure of their work, making it more easily readable and less confusing for the related audience.

Discussion and Conclusion should be separated into unique subsections.

Specific comments:

Table 4:Isn’t the training accuracy in RF derived as 1-OOB ? It certainly seems like that from the table. Therefore, it is quite redundant to present both the OOB error and OOB accuracy.

Author Response

Response to Reviewer 1

General comments

We thank the reviewer for his/her useful comments and suggestions. We have taken into consideration all the reviewer’s comments and suggestions in revising our manuscript. The following summarizes how we have responded to the reviewer’s comments and suggestions. Please note the reviewer’s comments and suggestions are italics, while our response is in regular font.

Response to specific comments
Comments and Suggestions for Authors

The authors assess the contribution of various inputs and transformations of S1/S2 data for a land cover application over a set of cities in Zimbabwe. The application is novel, and the authors clearly justify the rational behind it, aligning well with the UN SDG. The manuscript is generally well written, but it is tediously long. I believe some sections can be shortened and some material can be transferred to a supplementary appendix if the authors see them as critical.

For example, a lot of Sentinel information is perhaps unnecessary (there is no need to me at least, to present tables describing the sentinel bands, a proper citation should suffice).

We have shortened the manuscript according to the reviewer's comments. In addition, we removed table 2, which contained general Sentinel-2 information. We also revised the sentence as follows:

In this study, we selected nine spectral bands from S2 level-2A orthorectified atmospherically corrected surface reflectance imagery, which are commonly used for land cover mapping applications. The selected bands are band 2 (Blue), band 3 (Green), band 4 (Red), band 5 (Vegetation red edge) (VRE1), band 6 (VRE2), band 7 (VRE3), band 8 (Near infrared) (NIR), band 11 (Short-wave infrared) (SWIR1), and band 12 (SWIR2).

Moreover, although you could present the training accuracy for Harare as it is a more detailed section, I find no need to present whole sections for OOB accuracy/error results for the rest of the cities (could go in the supplementary material). The validation (testing accuracy) related results, should suffice. It is important to understand that the OOB accuracy is considered a pseudo-test metric – as it is an out of sample validation and some researchers use it as an alternative to an independent test. Using both sequentially, and deducing different conclusions from each one is confusing for the readers.

We removed the training accuracy for the other cities.

Overall, I commend the authors for the work done, and I certainly believe their work is worth publishing, upon some changes refining the structure of their work, making it more easily readable and less confusing for the related audience.

We have improved structure. For example, we improved the description of the accuracy assessment and simplified Table 4 for better interpretation. We also removed Figure 12 since the user’s accuracy and producer’s accuracy are now included in table 4.

Discussion and Conclusion should be separated into unique subsections.

We have separated Discussion and Conclusions according to the reviewer’s comments. The discussion is in section 5, while the conclusion is presented in section 6.

The results are presented in section 4, while discussions are provided in section 5. Finally, conclusions are presented in section 6.

Specific comments:

Table 4:Isn’t the training accuracy in RF derived as 1-OOB ? It certainly seems like that from the table. Therefore, it is quite redundant to present both the OOB error and OOB accuracy.

We provided both the OOB error and OOB accuracy for transparency. Furthermore, we want to show clearly how OOB error varies as explained in the manuscript.

Reviewer 2 Report

I read the paper with great interest.

The topic is interesting especially for people who classify satellite images. Classification work was well planned, done and described. The text is long but easy to read.

Abstract - an abstract is too long, over 300 words long, it should be shortened to 200 according to the letter's guidelines.

The first line under Fig. 1. “Bulawayo –“replace it with “Bulawayo”, the dash is not used for the other descriptions of the test sites.

Comments to Section 3.1.1 Satellite imagery

Instead of S-1 and S-2, please use S1 and S2, which are more common. Please also change the other abbreviations: instead of S-2RS use S2RS or S2-RS, proposed abbreviations are more readable.

Composites - add a sentence explaining how composites are prepared and a reference to literature.

Table 1:  please provide the registration year of images and their number for each set.

Fig 2.: What colour composition was used for S2 images, in the text is "false colour", but channels must be specified. In my opinion, you should write e.g. RGB (8,11,4)

Table 3.:  A column with class numbers should be added. Training sites per class are expressed by pixels or polygons?

Fig 5.:  Improve the signature, .. model for land cover classes ….

Comments to Section 4.1.2. Evaluation of land cover maps

Fig. 6c. small drawings are difficult to interpret and should be improved. On the level "0", the image is unreadable by the dense "o" symbols.

Fig. 11c: the drawing is "heavy" please make it similar to Fig. 12.

Please better describe “error-adjusted”. Function in rating evaluation is not commonly used, and references should be added. By doing it the contents of Table 5 will be better interpreted.

This paragraph should be better expressed, in this form is difficult to follow the idea:
 "Table 5 shows substantial difference between mapped area and error-adjusted map for the bare areas class, which indicates classification problems. Generally, there are small or no marginal differences in individual accuracies for cropland, woodland and grass / open areas classes. However, there is a substantial difference between the mapped area and error adjusted map for the cropland class, which suggests an overestimation of this class. For the water class, there are more errors of omission, which means that the RF classifier missed most of the water areas. "

Author Response

Response to Reviewer 2

General comments

We thank the reviewer for his/her useful comments and suggestions. We have taken into consideration all the reviewer’s comments and suggestions in revising our manuscript. The following summarizes how we have responded to the reviewer’s comments and suggestions. Please note the reviewer’s comments and suggestions are italics, while our response is in regular font.

Response to specific comments
Comments and Suggestions for Authors

I read the paper with great interest.

The topic is interesting especially for people who classify satellite images. Classification work was well planned, done and described. The text is long but easy to read.

We revised the text and improved some sections. First, we removed table 2, which contained Sentinel-2 information. Second, we removed the training accuracy for the other cities, which is just describing the training model accuracy. Third, we revised “Classification accuracy assessment” and also removed figure 12 (User’s accuracy and producer’s accuracy of estimated area proportion).

Abstract - an abstract is too long, over 300 words long, it should be shortened to 200 according to the letter's guidelines.

We have revised the abstract and shortened it according to the reviewer's suggestion.

The first line under Fig. 1. “Bulawayo –“replace it with “Bulawayo”, the dash is not used for the other descriptions of the test sites.

We revised the sentence according to the reviewer's suggestion.

Bulawayo is the second largest city in Zimbabwe, which is located in the southwestern part of Zimbabwe.

Comments to Section 3.1.1 Satellite imagery

Instead of S-1 and S-2, please use S1 and S2, which are more common. Please also change the other abbreviations: instead of S-2RS use S2RS or S2-RS, proposed abbreviations are more readable.

We changed the abbreviations to make it more readable as suggested by the reviewer.

Composites - add a sentence explaining how composites are prepared and a reference to literature.

We have added a sentence as suggested by the reviewer.

Note that S1 imagery is displayed in false color VV (red), VH (green), and VV (blue) for visualization purposes only, while S2 imagery is displayed in false color band 8 (red), band 4 (green), and band 3 (blue).

Table 1: please provide the registration year of images and their number for each set.

We revised table 1 and also included the year the images were acquired in the sentence below.

We derived seasonal Sentinel-1 (S1) and Sentinel-2 (S2) data for 2020 from Google Earth Engine [42] to map land cover in four major urban areas in Zimbabwe (Table 1).

Table 1. Summary of S1 and S2 data used in the study.

Compiled Imagery

Date Range

Season

Number of images/bands

Remarks

Mean and median S1

1 January - 30 March 2020

Rainy

4

IW swath mode 250 km,

VV and VH polarization,

pixel spacing (10) m, Ascending orbit

1 April - 30 June 2020

Post-rainy

4

1 July - 30 October 2020

Dry

4

Median S2

1 January - 30 March 2020

Rainy

9

Bands 2, 3, 4, 8 at 10 m spatial resolution;

Bands 5, 6, 7, 8a, 11 and 12 resampled to 10 m

1 April - 30 June 2020

Post-rainy

9

1 July - 30 October 2020

Dry

9

Fig 2.: What colour composition was used for S2 images, in the text is "false colour", but channels must be specified. In my opinion, you should write e.g. RGB (8,11,4)

We have specified S2 false color images as follows:

S2 imagery is displayed in false color RGB (8,4,3).

Table 3.: A column with class numbers should be added. Training sites per class are expressed by pixels or polygons?

Please note that Table 3 (formerly table 3) shows the number or training sites per class, which are all polygons as stated in the table.

Fig 5.: Improve the signature, .. model for land cover classes ….

We have revised the figure as:

Figure 5. Training model land cover class errors for Harare.

Comments to Section 4.1.2. Evaluation of land cover maps

Fig. 6c. small drawings are difficult to interpret and should be improved. On the level "0", the image is unreadable by the dense "o" symbols.

We removed the density plot, which is difficult to interpret and then focused only on the box plot, which is easier to understand. Below is the revised sentence.

Figure 6c shows box plots of backscatter (dB) derived from the training data. The overlap between most of the classes is clearly observed from the distributions, which vary significantly.

Fig. 11c: the drawing is "heavy" please make it similar to Fig. 12.

We have revised figure 11c according to the reviewer's suggestion.

Please better describe “error-adjusted”. Function in rating evaluation is not commonly used, and references should be added. By doing it the contents of Table 5 will be better interpreted.

We improved the description of the accuracy assessment and simplified Table 4 for better interpretation. We also removed Figure 12 since the user’s accuracy and producer’s accuracy are now included in table 4. Below are the additional sentences.

Note that the rigorous (unbiased) accuracy assessment is based on accuracy, which is calculated using area proportions not sample counts [58,59]. This means that absolute counts of the sample are converted into estimated area proportions using the equation provided by Olofsson et al. [59]. Therefore, the rigorous (unbiased) accuracy assessment results are reported in actual area units (km2 or hectares) and area proportions, which is more meaningful than mere pixel counts.

This paragraph should be better expressed, in this form is difficult to follow the idea:
"Table 5 shows substantial difference between mapped area and error-adjusted map for the bare areas class, which indicates classification problems. Generally, there are small or no marginal differences in individual accuracies for cropland, woodland and grass / open areas classes. However, there is a substantial difference between the mapped area and error adjusted map for the cropland class, which suggests an overestimation of this class. For the water class, there are more errors of omission, which means that the RF classifier missed most of the water areas.
"

We revised the paragraph as follows:

Note that the error matrix incorporates the standard error based on the total area of each land cover class. Therefore, the land cover class area estimates are not biased.

Table 4 shows the mapped land cover class area estimates, ± 95% confidence interval (CI), the user’s accuracy, and the producer’s accuracy. The accuracy assessment results show that overall accuracy is 72.5%, while the individual class accuracies vary significantly (Table 4). The user´s accuracy of estimated area proportion is higher than the producer´s accuracy of estimated area proportion for the built-up class. Therefore, there are more errors of omission given that some of the built-up areas were missed, especially in vegetated low density areas. The bare areas class has low individual accuracies, which indicates severe classification problems. This is attributed to the high class error for the bare areas class (Figure 5).
